# Sensitivity to the sources of uncertainties in the modeling of atmospheric CO$_2$ concentration within and in the vicinity of Paris city

Jinghui Lian[1,a], François-Marie Bréon[1], Grégoire Broquet[1], Thomas Lauvaux[1], Bo Zheng[1,b], Michel Ramonet[1], Irène Xueref-Remy[2,c], Simone Kotthaus[3], Martial Haeffelin[3] and Philippe Ciais[1]

[1]Laboratoire des Sciences du Climat et de l'Environnement (LSCE), IPSL, CEA-CNRS-UVSQ, Université Paris-Saclay, 91191 Gif sur Yvette Cedex, France

[2]Institut Méditerranéen de Biodiversité et d'Ecologie marine et continentale (IMBE), Aix Marseille Université, CNRS, IRD, Avignon Université, Aix-en-Provence, France

[3]Institut Pierre Simon Laplace (IPSL), CNRS, École Polytechnique, Institut Polytechnique de Paris, 91128 Palaiseau Cedex, France

[a]Now at: Suez Group, Tour CB21, 16 Place de l'Iris, 92040 Paris La Défense Cedex, France

[b] Now at: Institute of Environment and Ecology, Tsinghua Shenzhen International Graduate School, Tsinghua University, Shenzhen 518055, China

[c]formerly at: Laboratoire des Sciences du Climat et de l'Environnement (LSCE), IPSL, CEA-CNRS-UVSQ, Université Paris-Saclay, 91191 Gif sur Yvette Cedex, France

*Correspondence to:* Jinghui Lian (jinghui.lian@suez.com)

**Abstract**. The top-down atmospheric inversion method that couples atmospheric CO$_2$ observations with an atmospheric transport model has been used extensively to quantify CO$_2$ emissions from cities. However, the potential of the method is limited by several sources of misfits between the measured and modeled CO$_2$ that are of different origins than the targeted CO$_2$ emissions. This study investigates the critical sources of errors that can compromise the estimates of the city-scale emissions and identifies the signal of emissions that has to be filtered when doing inversions. A set of one-year forward simulations is carried out using the WRF-Chem model at a horizontal resolution of 1 km focusing on the Paris area with different anthropogenic emission inventories, physical parameterizations and CO$_2$ boundary conditions. The simulated CO$_2$ concentrations are compared with in situ observations from six continuous monitoring stations located within Paris and its vicinity. Results highlight large nighttime observation-model misfits, especially in winter within the city, which are attributed to large uncertainties in the diurnal profile of anthropogenic emissions as well as to errors in the vertical mixing near the surface in the WRF-Chem model. The nighttime biogenic respiration to the CO$_2$ concentration is a significant source of modeling errors during the growing season outside the city. When winds are from continental Europe and the CO$_2$ concentration of incoming air masses is influenced by remote emissions and large-scale biogenic fluxes, differences in the simulated CO$_2$ induced by the two different boundary conditions (CAMS and CarbonTracker) can be of up to 5 ppm. Nevertheless, our results demonstrate the potential of our optimal CO$_2$ atmospheric modeling system to be utilized in atmospheric inversions of CO$_2$ emissions over the Paris metropolitan area. We evaluated the model performances in terms of wind, vertical mixing and CO$_2$ model-data mismatches, and developed a filtering algorithm for outliers due to local contamination and unfavorable meteorological conditions. Analysis of model-observation misfit indicates that future inversions at the mesoscale should only use afternoon urban CO$_2$ measurements in winter and suburban measurements in summer. Finally, we determined that errors related to CO$_2$ boundary conditions can be overcome by including distant background observations to constrain the boundary inflow or by assimilating CO$_2$ gradients of upwind-downwind stations rather than by assimilating absolute CO$_2$ concentrations.

## 1 Introduction

Worldwide, almost two-thirds of global final energy consumption takes place in urban agglomeration areas that have a high population density and corresponding infrastructure, and cities directly release about 44 % of the global energy-related CO$_2$

emissions (IEA, 2016; Seto et al., 2014). Due to progressing urbanization processes, the number of people living in cities is expected to increase from the current 7.7 billion in 2019 to more than 9.7 billion by 2050 (United Nations, 2019). More than ever, cities are at the front line of climate change mitigation and take the lead in energy transition and emission reduction of greenhouse gases.

Currently, a variety of efforts are underway to quantify cities' total $CO_2$ emissions and establish a high spatially and temporally resolved emission inventory for supporting urban emission mitigation strategies. An independent monitoring of city emissions is highly desirable, which could be delivered by the top-down atmospheric inversion method using regional high-resolution transport models together with ground-based urban $CO_2$ concentration networks and/or satellites with imagery capabilities. The so-called atmospheric inversion provides an optimized estimate of $CO_2$ emissions aiming at the best agreement between atmospheric $CO_2$

measurements and their simulated equivalents. It relies on the filtering of the $CO_2$ signal associated with the urban emissions at the targeted spatial and temporal scales from other sources of misfits between measured and modeled $CO_2$ concentrations. These other sources of misfits include uncertainties in the atmospheric transport, in atmospheric $CO_2$ conditions that are used at the boundaries of the regional model, in the natural $CO_2$ fluxes within the modeling domain, but also in the spatial and temporal distribution of the urban emissions at scales finer than the targeted ones. Even when controlling the emissions at a relatively high temporal and

spatial resolution, city-scale inversion frameworks have generally targeted monthly to annual budgets of the emissions at the city scale, or for large areas of these cities (strong temporal and spatial correlations are assumed). The uncertainties in the assumed temporal and spatial emission variations induce a critical source of error poorly constrained by the inversions due to the lack of data (Bréon et al. 2015; Lauvaux et al. 2016). The spatial and temporal allocation of the emissions is generally derived from high-resolution gridded inventory based on uncertain activity data in the transportation, residential, and power sectors (Gately et al.,

2017). Moreover, local sources of $CO_2$ in the vicinity of an urban station can cause variations of atmospheric $CO_2$ that are not captured by the inventories and transport models of kilometric scale that have been used for city inversions so far (Boon et al., 2016; Lian et al., 2019). Further, cities have green areas and are surrounded by rural areas that actively take up $CO_2$ in the daytime during the growing season. Uncertainties and variability in those biogenic fluxes also significantly affect the results of atmospheric inversions (Hardiman et al., 2017).

Uncertainties in modeling the atmospheric transport of $CO_2$ are exacerbated in urban areas due to building obstacles that generate specific mixing processes and modify the wind speed and direction. In addition, sensible heat emissions at the surface of urban areas enhance vertical mixing, increase the depth of the boundary layer (Dupont et al., 1999) and can drive regional mesoscale circulations under certain conditions. To reduce transport uncertainties in inversions over urban areas, one can use dedicated urban surface schemes (e.g. Nehrkorn et al., 2013; Feng et al., 2016). More general approaches to reduce transport errors rely on the

assimilation of upper-air weather data or on the optimization of the model configuration, e.g. based on comparisons against independent wind measurements (e.g. Deng et al., 2017). But some errors remain difficult to quantify, such as those from local circulations and complex meteorological conditions (Martin et al., 2019). As a consequence, an empirical selection of the data to be assimilated is usually performed, which is more or less stringent depending on each urban station and transport model. Typical selection criteria of continuous urban $CO_2$ data consist of (i) using only measurements acquired during the afternoon when a well-

developed convective mixing layer is expected; (ii) using only observations when the wind speed is above a given threshold; (iii) removing statistical outliers.

Uncertainties in $CO_2$ boundary conditions arise from the fact that city-scale inversions are performed over a limited spatial domain that receives $CO_2$ signals from outside. These boundary conditions usually cannot be measured explicitly and they can be complex for continental cities that receive $CO_2$ advected by long-range and middle-range transport from other urban areas and biogenic

fluxes. Göckede et al. (2010) found that small biases in $CO_2$ boundary conditions could lead to large errors (~47%) in the posterior

annual state-level $CO_2$ fluxes of Oregon. Lauvaux et al. (2012) found that a 0.55 ppm bias of $CO_2$ boundary condition induced a 10% bias in the posterior annual $CO_2$ flux of Iowa and surrounding states. In order to try to eliminate the bias from boundary conditions, Bréon et al. (2015) and Staufer et al. (2016) proposed to assimilate $CO_2$ gradients between upwind-downwind stations in inversions of $CO_2$ fluxes of the Paris area, which reduces the number of data that can be assimilated.

Series of $CO_2$ transport and inverse modeling studies have been conducted for Paris (Bréon et al., 2015; Staufer et al., 2016; Wu et al., 2016; Broquet et al., 2018; Xueref-Remy et al., 2018). Since the year 2014, the Paris $CO_2$ monitoring network has been relocated and expanded with seven in-situ $CO_2$ stations combined with meteorological measurements. The present network, in particular the two newly built urban sites, is expected to provide new insights into the urban $CO_2$ characteristics. Lian et al. (2018) and Lian et al. (2019) attempted at setting up a high-resolution atmospheric transport modeling framework that is more robust or at least more flexible in terms of parameterization than those used in the previous Paris studies to account for the impacts of the urban effects, the biogenic flux and the model physics, which makes it promising to enlarge the set of data that can be assimilated for the inversions of the Paris $CO_2$ emissions, and in a more general way, to strengthen the inversions. Therefore, a full re-assessment of the modeling skills and of the main sources of misfits between the observations and the model is needed on these new bases. More specifically, we analyze in detail the model-measurement mismatches so as to identify critical sources of errors that would compromise a high-resolution atmospheric inversion of urban $CO_2$ emissions in the Paris area. A set of forward simulations of atmospheric $CO_2$ concentration are performed at 1-km horizontal resolution using the WRF-Chem model (Grell et al., 2005) with different anthropogenic emission inventories, physical parameterizations and $CO_2$ boundary conditions over the Paris for the 1-year period spanning December 2015 to November 2016. The main objectives of this paper are to provide a rigorous and detailed error characterization of our atmospheric modeling system and to determine the data selection method (i.e. filtering of short-term model errors and local contamination) and $CO_2$ boundary condition specifications at city scale during both daytime and nighttime over the full year period. We also address the question to what extent these model-measurement mismatches might be reduced and how our proposed diagnostics could be used to provide additional constraints for the inversion of $CO_2$ emissions at the city scale.

## 2 Methods

### 2.1 Experimental design

The WRF-Chem V3.9.1 model was used to simulate hourly atmospheric $CO_2$ concentrations over the Paris region. Details regarding the model setup and the reference data used in the simulations are outlined briefly below and described in Lian et al. (2019). The model was configured with one-way nesting of three modeling domains (D01, D02, and D03 in Figure 1a) at horizontal grid resolutions of 25, 5 and 1 km respectively, in which the innermost one (D03) covers the Île-de-France region (IdF, which is the administrative area that includes the Paris urban area) and its surrounding. The meteorological initial and lateral boundary conditions were retrieved from the global European Centre for Medium-Range Weather Forecasts (ECMWF) Interim Re-Analysis data (ERA-Interim) with 0.75°×0.75° horizontal resolution at 6-hourly update intervals (Berrisford et al., 2011). The grid nudging option in WRF to relax the model to ERA-Interim on large scales was applied to temperature and wind fields at model levels above the planetary boundary layer (PBL) of the outer two domains. We also used the surface analysis nudging and observation nudging options to assimilate the National Centers for Environmental Prediction (NCEP) operational global upper-air (ds351.0) and surface (ds461.0) observation weather station data (https://rda.ucar.edu/datasets/ds351.0/; https://rda.ucar.edu/datasets/ds461.0/), which are described in more detail in Lian et al. (2018). The biogenic $CO_2$ fluxes were calculated online in WRF-Chem by the diagnostic biosphere Vegetation Photosynthesis and Respiration Model (VPRM) (Mahadevan et al., 2008; Ahmadov et al., 2007, 2009). The

values of the four parameters (α, β, λ, and PAR0) for each vegetation category used by VPRM have been optimized against eddy covariance flux measurements over Europe collected during the Integrated Project CarboEurope-IP (http://www.carboeurope.org/).

### 2.1.1 Atmospheric physics options

An accurate physical parameterization of atmospheric transport model is critical to numerical simulations of the meteorology and $CO_2$ concentrations within and around urban areas. A set of numerical experiments was performed to assess the sensitivity of the simulations with the WRF-Chem model to the choice of different PBL and urban canopy schemes. These two physics schemes were selected as they have a more significant impact on the simulated meteorological variables than the other schemes based on our previous sensitivity study (Lian et al., 2018; Lian et al., 2019), and thus the differences between simulations with these two physical options could provide an estimate of the atmospheric transport uncertainty over the Paris region. The characteristics of $CO_2$ distributions are highly related to the PBL structure and its temporal evolution. We carried out sensitivity experiments with three different PBL parameterization schemes (Table 1a), including the Yonsei University scheme (YSU) (Hong et al., 2006), the Mellor-Yamada-Janjic scheme (MYJ) (Janjić, 1990, 1994), and the Bougeault-Lacarrère scheme (BouLac) (Bougeault and Lacarrere, 1989). In addition, two different urban surface parameterizations were investigated, the single-layer urban canopy model (UCM) (Chen et al., 2011) and the multilayer urban canopy model BEP (Building Effect Parameterization) (Martilli et al., 2002) (Table 1a). The non-local YSU scheme was used with the Revised MM5 Monin-Obukhov surface layer scheme (Jiménez et al., 2012), whereas the two local MYJ and BouLac schemes were used with the Monin-Obukhov Eta Similarity surface layer scheme (Janjić, 1996). All other physics options were identical for all sensitivity runs: WSM6 microphysics scheme (Hong and Lim, 2006), RRTM longwave radiation scheme (Mlawer et al., 1997), Dudhia shortwave radiation scheme (Dudhia, 1989), Unified Noah land-surface scheme for non-urban land cover surface energy fluxes (Chen and Dudhia, 2001). The Grell 3D ensemble cumulus convection scheme (Grell and Dévényi, 2002) was only employed for the outer domain (D01). These options correspond to those selected by Lian et al. (2018) which showed good performances for simulating near-surface winds and temperatures over the Paris region. The simulations were performed for a period of 15 months from September 2015 to November 2016 including a spin-up of three months.

### 2.1.2 Anthropogenic emission inventories

Numerical experiments were carried out to assess the modeled $CO_2$ sensitivity to the use of different anthropogenic emission maps and to get insights on the signature of typical uncertainties in such maps (Table 1b). The two spatially and temporally explicit emission fields derived from inventories used in this study were the 2010 AirParif inventory at a spatial resolution of 1 km (AIRPARIF, 2013) and the European greenhouse gas emission inventory (5 km × 5 km resolution) for the base year 2005 developed by the Institute of Economics and the Rational Use of Energy (IER), University of Stuttgart (http://carboeurope.ier.uni-stuttgart.de/). Both inventories simulated monthly, weekly and diurnal profiles and were rescaled on the basis of the ratios of the national annual budgets of $CO_2$ emissions for the countries within the domain, between the base year and the year of simulation (2015/2016), taken from Le Quéré et al. (2018).

Figures 1b and 1c show a map of daily $CO_2$ emission within the IdF region for a weekday in November 2015 from the 1-km AirParif inventory and the 5-km IER inventory respectively. The figures show that the emissions are the largest within and in the near vicinity of the Paris administrative city (the core of the urban area). The suburban area extends approximately 15 km outside of the city limits. The AirParif inventory is expected to offer a more robust description of the emissions for the year of simulation than the IER inventory does because it uses more local and more recent data (AIRPARIF, 2013).

The temporal variations of emissions also show some differences, in particular when differentiated per sector (Figure 2). The emissions, split up by five sectors (namely building, surface traffic, energy, industry, and all other sectors), are different both in terms of magnitude and diurnal cycle between the two inventories. This is true both for the very center of Paris where the CDS $CO_2$ measurement station is located (Figure 2a) and on a relatively large (5-km) spatial scale (Figure 2b). The relative difference between the two inventories is smaller in terms of total emissions. Figure 2 also shows the total emission for both IER and AirParif inventories as a function of time in the day. At the larger scale (Figure 2b), a substantial difference is found during the early morning when AirParif shows emissions that are much smaller than those of IER, and with a clear temporal trend.

In order to investigate the impact of the spatio-temporal distribution (especially the prescribed diurnal profile) of emissions on the modeled $CO_2$ concentrations, we made a one-month simulation using these two anthropogenic inventories together with their respective temporal profiles (Table 1b). Within the same group of simulations, two more sensitivity tests of the diurnal profile were also carried out by using: (i) a constant temporal profile (each pixel has a different emission, but constant in time based on the temporal average of the AirParif inventory) and (ii) a constant and spatially homogeneous emission where the emissions are distributed uniformly over the IdF whole territory. Distinct $CO_2$ tracers are used for each of the four experiments to quantify their respective impacts on the atmospheric $CO_2$ concentration, for a given configuration of the WRF-Chem model. The simulation was carried out for the one-month period of January 2016 when the influence of regional biogenic flux on $CO_2$ signals is relatively small compared to that of anthropogenic flux.

### 2.1.3 Boundary conditions for $CO_2$

A set of sensitivity experiments was designed to investigate the impact of different $CO_2$ boundary conditions on the Paris $CO_2$ concentrations (Table 1c). The initial and lateral boundary conditions for $CO_2$ concentration fields used in the sensitivity experiments were respectively taken from two global $CO_2$ atmospheric inversion products at 3-hourly update intervals: CAMS and CarbonTracker. CAMS has a horizontal resolution of $3.75° \times 1.90°$ (longitude $\times$ latitude), with 39 hybrid layers in the vertical (version v16r1, https://apps.ecmwf.int/datasets/data/cams-ghg-inversions/; Chevallier, 2017a, 2017b). CarbonTracker has a horizontal resolution of 3° in longitude and 2° in latitude, with 25 vertical layers (version CT2017, http://carbontracker.noaa.gov; Peters et al. 2007). Both global datasets were interpolated onto the outermost domain of WRF-Chem (D01) (bilinearly in longitude, longitude and linearly in pressure) so as to provide the lateral boundary conditions for $CO_2$ simulations. Given that CarbonTracker has an averaged value over each 3-hourly interval (the times on the date axis are the centers of each averaging period), it was also linearly interpolated in time to ensure consistency with both CAMS and the interval of input data for WRF-Chem (e.g. the value at 00 UTC was generated by interpolating the one at 22.5 UTC of the previous day with the one at 1.5 UTC of the same day).

Figure 3 shows time series of average differences in $CO_2$ concentration between CAMS and CarbonTracker at each of the four lateral boundaries, averaged over the lowest 0.7 km above ground level (AGL), of D01 for both 00 UTC and 12 UTC. These time series are the spatial mean and standard deviation ($\pm 1\sigma$) over each boundary (a latitudinal transect for western and eastern boundaries / a longitudinal transect for southern and northern boundaries). In general, winds blow mostly from the west in all seasons over the domain of interest. Small differences at the western boundary are observed under the influence of prevailing westerlies with annual means of the spatial mean and standard deviation of $0.01 \pm 2.8$ ppm for 00 UTC and $0.4 \pm 1.8$ ppm for 12 UTC, which is expected as the air masses are advected from clean air (oceanic) areas. In contrast, the differences are significantly larger at the eastern boundary ($-4.8 \pm 7.4$ ppm for 00 UTC and $-1.7 \pm 3.3$ ppm for 12 UTC), but can vary from day to day depending on the synoptic weather condition. A possible explanation could be that both fossil fuel and biogenic $CO_2$ fluxes and associated uncertainties are larger over the European continent than over the oceans. It may also be caused by the sensitivity of the modeled $CO_2$ concentrations to the transport fields over the Alps mountain region at the eastern boundary. This feature indicates that CAMS

and CarbonTracker may provide substantially different continental $CO_2$ background signals to the inner domain when the wind blows from the east. Moreover, the magnitude and variability of the differences are overall smaller at noon compared to those at midnight. The variability of nighttime differences appears relatively larger in summer than those in winter. Note that the $CO_2$ differences between CAMS and CarbonTracker are much smaller for the upper layers above 0.7 km AGL, with annual means of the spatial mean and standard deviation of -0.4 ± 0.4 ppm for both 00 UTC and 12 UTC at the eastern boundary (Figure S1).

The WRF-Chem simulation with boundary conditions from CarbonTracker used the same physics schemes and prior fluxes as the one with boundary conditions from CAMS (also defined as the control run), whereas it was only carried out for the parent domain (D01) without nesting over a full-year period (2015.09-2016.11). The simulation was restarted every 5 days with the $CO_2$ initial values from the previous run. Given the fact that lateral boundary conditions are fed to the nested domain from the parent (the nest is driven along its lateral boundaries by the parent domain), results from D01 should therefore be representative enough to access the modeled $CO_2$ sensitivity over the IdF region to the use of different $CO_2$ boundary conditions.

### 2.2 $CO_2$ in situ and meteorological observations

For the model evaluation, we use observations from six in situ continuous $CO_2$ monitoring stations established in the IdF region. Four stations (AND, COU, OVS, SAC) are located within peri-urban areas and two (JUS and CDS) are located within the Paris city. The SAC station has two air inlets placed at 15 m and 100 m AGL respectively. Each of the other stations is equipped with a continuous $CO_2$ gas analyzer and inlets located on rooftops or on towers with heights varying from 20 m to 60 m AGL. The $CO_2$ analyzers are high-precision cavity ring-down spectroscopy instruments with a calibration system using three reference gases tied to the WMO $CO_2$ X2007 scale every 1 to 6 months (Tans et al., 2011). The six stations within IdF are complemented by two ICOS atmospheric background $CO_2$ tall tower monitoring stations (TRN and OPE) located respectively 101 km and 235 km away from the center of Paris. In this study, observations from these two stations are only used as background sites and to provide additional support and validations for the results of diagnostics made at the SAC site. In addition to the $CO_2$ measurements, the hourly air temperature, wind speed and wind direction are also measured at a height of 100 meters above ground level at the SAC station. Let us remind that the meteorological data at the SAC station are not included in the data assimilation process of the NCEP operational global weather observation subsets used in the WRF nudging program (section 2.1). They could therefore be considered as independent observations for the evaluation of the model performance in simulating the meteorological fields. The PBL heights are obtained from profile measurements of a Lufft CHM15k ceilometer operated at the SIRTA site (Haeffelin et al., 2005) located about 20 km southwest of Paris center. The PBL heights derived using the STRATfinder algorithm are most reliable in the afternoon during considerable convection while the detection of shallow layer heights below 300 m (e.g. at night or cold seasons) is associated with increased uncertainty (Kotthaus et al., 2020). The locations of all the observing stations together with their sampling heights are shown in Figure 1.

### 3 Results

### 3.1 Overall model performance

In this section, we start with an evaluation of the overall performances of the control run (BEP_MYJ) in simulating both meteorological fields and atmospheric $CO_2$ over the full-year period from December 2015 to November 2016.

### 3.1.1 Meteorological fields

Since the accuracy of the modeled $CO_2$ concentrations depends on the quality of the meteorological model, the simulated meteorology by WRF was first evaluated against observations at SAC100 and SIRTA stations with a focus on three variables (air temperature, wind and PBL height). Figure 4 shows the time series of the 1-year daily afternoon mean (11-16 UTC) observed and modeled temperature, wind speed and wind direction at SAC100 station, together with their statistics summarized in the scatter plots. The daily nighttime mean (21-05 UTC) data are shown in Figure S2. In general, both daytime and nighttime temperature are well reproduced by WRF with correlation coefficient, RMSE and MBE of 1.0, 0.44°C, 0.06°C and 0.99, 0.67°C, 0.23°C respectively. The analysis of the MBE shows that the wind speeds are slightly overestimated by WRF, with a bias of 0.96 m/s for afternoon and 0.68 m/s at night. As for the wind direction, the model-data misfits decrease with the increasing wind speed. Seasonal (and even some day-to-day) variations in the afternoon average PBL heights diagnosed from the model data are in general agreement with the observations at the suburban SIRTA site with a RMSE of 359 m and a positive bias of 82 m. Some disagreements between the model-data PBL height estimates can be expected given layer heights from aerosol-based methods (as here applied to the observations) tend to lag behind those determined from thermodynamic methods (applied to the model data) during the course of the day (Kotthaus et al., 2018). Relative agreement between PBL heights is reduced at night (Figure S2), as uncertainties are higher in both the observed layer heights (Section 2.2) and those diagnosed from the model data (Shin and Hong, 2011). In general, results in Figure 4 and Figure S2 show that the simulated meteorological fields agree reasonably well with observations both during day and night which indicates parameter settings suitable overall.

### 3.1.2 CO₂ concentration

The accuracy of model $CO_2$ estimates at the six in situ measurement stations is assessed using three statistical indicators corresponding to the hourly values: the correlation coefficient (R), the root mean square error (RMSE) and the mean bias error (MBE). We also use the K-nearest neighbors (KNN) algorithm with an outlier fraction of 0.1 (10%) to detect the largest model-observation mismatches so as to minimize their influences on the statistical results (Ramaswamy et al., 2000; Zhao et al., 2019). These large model-observation discrepancies are supposed to be due e.g. to the occasional contaminations from local sources of $CO_2$ emissions near the measurement station that cannot be resolved by the 1-km resolution model, or to the failure of the model in the description of $CO_2$ concentrations under some meteorological conditions such as heavy rains and storms, thick clouds with a thermodynamically stable inversion (diagnosed hereafter from Bulletin Climatique Météo-France, 2016). We further analyzed the filtered hourly concentrations (detailed in supplement material Figure S3 and S4) and confirmed the contamination at one of our sites (OVS) and the relationship between meteorological conditions and excluded modeled concentrations.

Figure 5 shows, for the six monitoring sites, the scatter plots of the BEP_MYJ simulated vs. observed all hourly $CO_2$ concentrations from December 2015 to November 2016. The typical $CO_2$ concentrations vary between 390 and 430 ppm, and up to 440 ppm within the city (JUS and CDS stations). For short periods, the concentrations can be much higher (both for the observations and the model), in particular within the city where values of more than 500 ppm are sometimes observed. However, these data are considered as "outliers" by the KNN. When considering all data-points, the correlations vary between 0.5 for the JUS station at the very center of Paris to 0.76 at AND. The correlations get larger (0.55-0.83) when KNN outliers are removed. This correlation is partly driven by the seasonal cycle of $CO_2$ and does not provide specific information on the model's ability to reproduce short-term variations. The MBEs are of a few ppm, and at the OVS station can go up to -6 ppm. The KNN removal of outliers tends to reduce these biases, but not so within the Paris city. Finally, the RMSEs are larger than 10 ppm in most stations. RMSE values are significantly reduced through the KNN data selection but nevertheless range between approximately 5 and 10 ppm.

Figure 6 shows the average diurnal cycles split up by season for the measurements and the model results. It also shows the corresponding differences between the model and the data. These figures clearly show the seasonal variability with summer concentrations smaller than those during the rest of the year (due to photosynthetic absorption), larger concentrations within the city (JUS and CDS) (in the largest cluster of anthropogenic emissions), and the strong diurnal cycle that is mostly driven by atmospheric mixing. Both observations and the model show a double-peak pattern in the diurnal cycle of $CO_2$ concentrations at the two urban stations in winter, concomitant with traffic peaks. In addition to the mean seasonal cycle that is generated by large-scale (continental, hemispheric) vegetation photosynthetic uptake and respiration coupled to long-range transport, the variability of the synoptic-scale atmospheric flow also impacts the seasonal concentrations. It is worth noting that the $CO_2$ concentrations in autumn 2016 (SON) are on average higher than the other seasons, even slightly larger than those of the winter period (DJF) in 2015. This is interpreted as the consequence of persistent anticyclonic conditions leading to dry and calm weather over the north of France, and thus $CO_2$ accumulation near the surface throughout that period (Bulletin Climatique Météo-France, 2016).

The model reproduces the main features of the average diurnal cycle of $CO_2$ during the different seasons but the measurement-model discrepancies can be significant. At noon and during the afternoon, the mean differences are on the order of a ppm (with the exception of JUS). The model underestimates $CO_2$ with a time-varying bias roughly ranging from 0 to 12 ppm across stations for all seasons during the night until around 05 UTC. The two stations within the city have different behaviors, with larger differences, in particular the JUS station located at the city center. For this station, significant measurement-model discrepancies varying from 0 to 7 ppm over time are found, even during the afternoon when a good agreement is found at the other stations. Moreover, the model reproduces much smaller amplitudes of $CO_2$ diurnal cycle than the observations, in particular at the suburban stations.

The measurements themselves have an accuracy that is on the order of a fraction of a ppm (Xueref-Remy et al., 2018) and measurement errors are therefore negligible when analyzing such model-data differences. In the following, we analyze in further detail the measurement-model discrepancies and attempt to identify cases when they appear to be mainly driven by uncertainties in the anthropogenic emissions, in the biogenic fluxes, in the physical parameterizations of the atmospheric transport model, or in the $CO_2$ boundary conditions at the limits of the atmospheric transport model.

## 3.2 Contribution of main sources of errors in the simulated $CO_2$ related to different factors

### 3.2.1 Emission inventory

The main objective when measuring $CO_2$ concentrations within or in proximity to the city is to estimate the anthropogenic emissions by means of an atmospheric inversion. It is then natural to seek, in the time series, unambiguous signatures of erroneous assumptions on the anthropogenic emissions. This is a difficult task as significant uncertainties in the atmospheric transport also impact the modeling results, while there is no knowledge of both "true" emissions and "true" transport.

Figure 7 shows the results of the sensitivity experiments that used different temporal profiles and spatial distributions of anthropogenic $CO_2$ emissions (see Table 1b and Figure 2). The decomposition of the $CO_2$ concentration per tracer makes it very clear that in January when the biogenic flux is small the diurnal cycle of $CO_2$ at the six measurement sites is almost entirely associated with the signature of the anthropogenic emissions. This may not hold during the growing season. None of the simulations show a diurnal cycle that is close to the observed one. The most striking error is the evolution of the concentrations through the night when observation show an increase in $CO_2$ while the model shows a clear decrease. Results in Figure 7c and 7d show that the impacts of biogenic flux and background condition on this simulated decrease are relatively small as they are on the order of a fraction of a ppm. Instead, the decrease of anthropogenic emissions during the night (Figure 2) explains part of the decrease in modelled concentrations. Assuming the AirParif inventory with a constant temporal profile, the decreasing trend at night is reduced and the modeled value (green line in Figure 7) is closer to the observation than the control run (BEP_MYJ). Further analysis shows

that the nighttime trend of the anthropogenic emission in January is mostly linked to residential heating in the inventory. The diurnal profile used for heating emissions in the AirParif inventory (with a significant decrease through the night) can thus be questioned.

Although there is a strong indication that the nighttime profile of the AirParif $CO_2$ emissions is erroneous and that heating emissions do not reduce strongly during the night, this error does not entirely explain the model-data misfit at CDS shown in Figure 7. This is proven by the fact that even the "constant emission" simulation does not reproduce the increasing concentration during the night. This implies that errors in atmospheric transport are also contributing to the model-data misfit, in particular concerning the vertical mixing near the surface. Further evidence for the transport deficiency is that the underestimations of nighttime $CO_2$ concentration are not only large at the two urban sites but also obvious at all rural stations (Figure 6).

To gain insights the impact of vertical transport, we show in Figure 8a the vertical distribution of the BEP_MYJ modeled $CO_2$ concentrations at CDS in January, together with time series of observed and simulated PBL heights at SIRTA. The modeled PBL heights are diagnosed using the 1.5-theta-increase method which defines the height of PBL as the level at which the potential temperature first exceeds the minimum potential temperature within the boundary layer by 1.5 K (Nielsen-Gammon et al., 2008). Results show that the model reproduces large vertical gradients in $CO_2$ concentrations in the low atmosphere levels, i.e. up to approximately 300 m AGL but mostly in the first 100 m. The largest concentrations are observed in low-wind speed conditions and when the PBL is shallow (Figure 8b). It is worth noting that the modeled $CO_2$ concentrations within the PBL are not vertically homogeneous but exhibit a strong gradient. This indicates that when the measurements are under a strong influence of upwind emissions or close to the large sources of emissions, the mixing is far from complete, even during the afternoon.

Moreover, both the BEP_MYJ and BEP_MYJ_IER model slightly overestimate $CO_2$ concentrations at CDS in the late afternoon and early evening (from 18 UTC to 22 UTC) not only in January (Figure 7) but also over the full year (Figure 6). This is interpreted as the consequence of a shift from a situation with convective mixing to stable nocturnal conditions around sunset occurring too early in the model. It may also be linked to an increase in traffic emissions during the evening rush hour, which could also lead to the overestimated modeled concentrations in the late afternoon.

### 3.2.2 Biogenic fluxes

To analyze the influence of biogenic fluxes on the $CO_2$ concentrations, we computed $CO_2$ horizontal differences between two sites (i) CDS, that is within the limits of the Paris city where the diurnal cycle in winter is dominated by anthropogenic emissions (see Figure 7) and (ii) SAC that is over a more rural area with a mix of crops and forest, so that the variations of $CO_2$ concentrations at that site are mostly driven by biogenic fluxes in the domain and $CO_2$ background conditions. Figures 9a and 9b show the time series of the observed and BEP_MYJ simulated horizontal differences in near-surface daily $CO_2$ concentration between CDS and SAC for two different periods of the day, the afternoon mean (11-16 UTC) and the nighttime mean (21-05 UTC) from December 2015 to November 2016.

The separate tracers from the WRF-Chem model make it possible to quantify the respective contribution of anthropogenic, biogenic and background sources to the $CO_2$ difference between CDS and SAC (Figure 9c and 9d). During the afternoon, the $CO_2$ differences are mostly positive and result primarily from the larger contribution of the anthropogenic emissions at CDS, both during the growing and non-growing season. This result indicates that the magnitude of daytime net carbon uptake plants between the stations does not fully offset that of the anthropogenic emissions, and thus the $CO_2$ concentration gradients between the upwind and downwind stations that are used in previous inversion studies can also be used even during the growing season (Bréon et al., 2015; Staufer et al., 2016). Nevertheless, the biogenic contribution to the gradient is not negligible with a potential impact on the estimate of the anthropogenic emissions from the measured gradient. During the night, there is a large measurement-model discrepancy

from June to September (unfortunately the SAC station had measurement gaps from May 3rd to June 23rd and from July 7th to July 12th). During this growing season period, the observed difference between CDS and SAC is negative at night (higher concentrations at SAC than at CDS), while the simulated difference is positive resulting from a large positive anthropogenic contribution and a smaller negative biogenic contribution. Figure S5 shows that this nighttime misfit between the modeled and observed $CO_2$ differences has a seasonal trend that follows closely the one of the modeled gross primary production (GPP). A large fraction of GPP realized each day is respired at night by plant maintenance respiration. The seasonal trend of the nighttime misfit between CDS and SAC thus indicates that the model underestimates plant respiration at night, and thus possibly GPP in the day. Although it is impossible to negate other hypotheses related to the atmospheric transport and vertical mixing, this result suggests that modeling nighttime $CO_2$ at rural stations is affected by systematic errors of respiration during the growing season, so that nighttime rural $CO_2$ data over that period can hardly be used in atmospheric inversions for inferring anthropogenic emissions.

Further insight on the $CO_2$ concentration dynamics at SAC is provided by the vertical differences that are derived from the measurements at two levels, 15 m and 100 m AGL, on a tall tower at that location (Figure 10). During the afternoon, the differences are small and there is little agreement between the observations and the simulated values (Figure 10a). This systematic bias between the observed and simulated $CO_2$ vertical gradients could be explained by an underestimation of the photosynthetic uptake. The vertical $CO_2$ differences are much larger at night with a fair agreement between the measurements and the simulated values in wintertime (Figure 10b). Although the nighttime time series show strong similitudes, there is a significant bias between the observations and the model during the growing season, but not so during the non-growing season. The seasonal phase of the vertical misfit is well correlated with the one obtained from the horizontal diagnostics, which tends to indicate the same bias in the estimated nighttime respiration.

The analyses of both Figure 9 and 10, together with similar results observed at other stations (Figure S6: e.g. the horizontal difference between CDS and COU, and the vertical difference at TRN), are consistent with the hypothesis that the respiration emission at night is underestimated by the VPRM model. If this nighttime respiration bias would be correlated with the daytime respiration bias (Reichstein et al., 2005), it would imply that modeled positive gradients of $CO_2$ between urban and rural stations could be overestimated during the growing season. We thus recommend for an inversion to control separately (with a priori) anthropogenic emissions and net ecosystem exchange, or even photosynthesis and respiration if additional data confirm a bias of respiration in VPRM.

### 3.2.3 Atmospheric transport

Uncertainty in simulated $CO_2$ due to transport errors can be evaluated empirically through the spread of simulated $CO_2$ by sensitivity experiments with different physical configurations of WRF-Chem. We have made five sensitivity simulations using the same surface fluxes and boundary conditions, but with three PBL schemes and two urban canopy schemes (see Table 1a).

Figure 11a shows the horizontal distribution of the monthly median standard deviation of simulated hourly $CO_2$ concentrations at approximately 20 m AGL using different physics schemes for two periods of the day (afternoon 11-16 UTC, nighttime 00-05 UTC), and for two months (January, July 2016). During January, the simulated $CO_2$ concentrations within the city, both in afternoon and nighttime, are highly sensitive to the choice of the physics scheme, with median standard deviations larger than 6 ppm. In contrast, the choice of the physics scheme has less influence on simulated $CO_2$ concentrations over suburban and rural areas in winter, with the median standard deviations of 1.2 ppm in the afternoon and 2 ppm at night. During the summer period, the smallest uncertainty of simulated $CO_2$ concentration resulting from different physics schemes is found in the afternoon with median standard deviations that are less than 1 ppm, which indicates that the various schemes provide very similar values. However, it is necessary to compare these standard deviations to the amplitudes of the anthropogenic emission signature. Indeed, the

anthropogenic signal may be understood as the "signal" for the estimate of the emission, while the spread of the five sensitivity simulations provides an indicator of the atmospheric transport uncertainty. We thus calculated the median ratios of the simulated anthropogenic $CO_2$ concentration (average over the five sensitivity runs) to its respective standard deviation of the total $CO_2$ signals among the five sensitivity runs, which we define as the "signal-to-noise ratio" (Figure 11b). The largest signal-to-noise ratio is found in the afternoon of summer within the urban area, indicating that the link between the anthropogenic emission and the $CO_2$ concentration can be derived from the model with the highest confidence for these conditions. However, during the summer, the nighttime $CO_2$ measurements over the suburbs are poorly suited for the inversion since the simulated $CO_2$ are highly sensitive to the choice of physics scheme and the signal-to-noise ratios are then relatively small ($< 1$).

Figure 11c shows the vertical distribution along a south-north transect through the JUS station in a similar way as Figure 11a. In general, the simulations with various physics options show very large variations in the modeled $CO_2$ concentrations (up to 7.5 ppm standard deviation) close to the surface, a few tens of meters above the emissions. The differences become much smaller (less than 1 ppm) with increasing altitude. This may be due to the fact that different physics schemes lead to different vertical mixing efficiencies, which has a strong impact on the vertical structure of $CO_2$ concentrations. Given that the measurements are acquired at a level where the vertical gradient is large and variable, it may also indicate that the measurement-model discrepancy is highly dependent on the physics parameterization in the representation of the vertical mixing process in near-surface layers. During the winter period, there is a considerable difference in the vertical concentration profiles reproduced by different physics schemes within the city, with the uncertainty extending to a higher altitude in the afternoon than those in the nighttime. Further away from the urban area, anthropogenic emissions are substantially lower, and the vertical gradient of $CO_2$ generated by the strong city emissions is smoothed out by the atmospheric convection and diffusion processes. As a consequence, much less uncertainty is associated with the choice of the physics scheme in the suburbs at altitudes above ~200 m AGL. As for the signal-to-noise ratio shown in Figure 11d, the large values within the city tend to indicate that the urban $CO_2$ data are well suited for an estimate of the emissions using the atmospheric inversion method.

We also accessed the respective contributions of anthropogenic and biogenic fluxes to the simulated spread of $CO_2$ concentrations using different physics schemes. This allows an estimate of the impact of uncertainties in the atmospheric transport modeling along with that of the impact of the various flux contributions. Figure 12 shows the statistics of the differences in simulated anthropogenic and biogenic $CO_2$ at approximately 20 m AGL between the control run (BEP_MYJ) and each of the other four sensitivity runs. The results in this figure are presented with the consideration of (i) two periods of the day (afternoon 11-16 UTC, nighttime 00-05 UTC); (ii) two months (January, July 2016); and (iii) three land use types (urban, crop and the others). Urban (7.4%) and crop (84.6%) are the two dominant land use types of the innermost model domain (D03) from the MODIS land cover database used in the WRF-Chem model, where the percentages in parenthesis indicate the proportion of each land use category to the total area. The other land use types (8.0%) mainly include grass, shrub, mixed forest, deciduous forest and evergreen forest. During the winter, the simulated anthropogenic $CO_2$ concentrations over the urban area are sensitive to the choice of the urban canopy scheme used in WRF-Chem, which is characterized by a substantial decrease in standard deviation from UCM to BEP (Figure 12a). The three simulations using the UCM scheme tend to produce higher anthropogenic $CO_2$ concentrations together with larger standard deviations with respect to the control run using the BEP scheme. This is because the BEP scheme generates more mixing in the lowest atmosphere especially from 07 to 14 UTC in the day and in winter relative to summer, which reduces the vertical gradient and therefore the largest concentrations near the surface (Figure S7). The two urban canopy schemes (UCM, BEP) show small differences in the simulation of anthropogenic $CO_2$ concentrations over the rural vegetated area for both seasons. This indicates that the choice of an urban canopy scheme is critical for simulating atmospheric transport at urban stations, but that the transport errors, without such scheme, remain mainly 'local' and have little remote influence at rural sites. That is, the choice of an urban

scheme impacts $CO_2$ concentrations over the urban areas but its impact on the larger scale transport is not significant enough to affect the simulated concentrations over rural areas. During the summer period, our results show that the modeled nighttime $CO_2$ concentrations are strongly sensitive to both the urban canopy and PBL schemes. This conclusion applies to both the urban and the rural areas.

Here, we quantify the uncertainty in the modeling results that is linked to the three PBL schemes and two urban canopy schemes. Clearly, there are other potential sources of atmospheric transport uncertainties that are not accounted for in this study. The simulated $CO_2$ differences among the ensemble of physics schemes tested here are therefore only a fraction of the full magnitude of model uncertainty. Nevertheless, this uncertainty is, in some cases, of similar magnitude as the measurement-model differences that have been shown in section 3.1.

**3.2.4 Boundary condition**

To investigate the uncertainty in $CO_2$ boundary conditions, we examined the modeled $CO_2$ sensitivity over the Paris region to the use of two different global $CO_2$ atmospheric inversion products as initial and boundary conditions for WRF-Chem (see Table 1c). Figure 13 shows all hourly $CO_2$ concentration differences between BEP_MYJ and BEP_MYJ_CT that used $CO_2$ fields from CAMS and CarbonTracker products respectively. The comparison is based on the simulated $CO_2$ in the 25-km grid cell of the outermost
domain (D01) containing the Paris city. For most time of the year (~73%), the differences in simulated $CO_2$ concentrations over Paris are within the range of ±1 ppm since they are mainly affected by the relatively low differences between CAMS and CarbonTracker at the western boundary of D01 under the influence of west winds (cf Figure 3). Nevertheless, considerable differences (up to 5 ppm) are observed during several synoptic episodes, which illustrates the magnitude of uncertainties linked with the boundary condition hypothesis. These magnitudes are similar to those of the impacts of different physics schemes on
simulated $CO_2$ concentrations over suburban and rural areas as shown in section 3.2.3. Under such circumstances, it requires the use of additional observations to constrain the boundary inflow in inversions. On the other hand, as the IdF region is exposed to a relatively well-mixed background atmosphere after a long-range transport of $CO_2$ from remote sources and sinks, one may expect that the resulting $CO_2$ concentration features are large scales. As a consequence, the potential modeling error induced by an erroneous boundary will be similar for monitoring stations located within Paris and its vicinity. This characteristic suggests that
the assimilation of upwind-downwind gradient in $CO_2$ concentrations in the inversion of city-scale emissions as done in previous studies could also be an effective way to minimize the potential biases both from the boundary conditions and from remote fluxes within the domain but outside the city (Bréon et al., 2015; Staufer et al., 2016).

**4 Conclusions and discussions**

We have analyzed $CO_2$ concentrations measured and modeled at six stations located within and in the surrounding of the Paris city.
Our objective was to identify the main causes of the $CO_2$ differences between the measurements and their simulated counterparts, with the overall goal to improve the quantification of anthropogenic emissions. To accomplish this, we have performed an ensemble-based sensitivity study and a full analysis of the uncertainties linked to anthropogenic inventories, biogenic fluxes, atmospheric transport and boundary conditions, either focusing on limited periods at different seasons or looking at the full-year period.

A preliminary identification of the modeling errors was first conducted with the KNN algorithm to identify the largest mismatches between the observations and the model results. These large discrepancies are most likely either related to specific measurement contaminations from local unresolved sources of $CO_2$ emissions, or to the model's inability to properly simulate the atmospheric

transport under specific meteorological conditions such as heavy rains and storms, thick clouds with a thermodynamically stable inversion. We should also note that removing outliers based on statistical analysis without attributing them to a real data contamination or model limitation has potential for data loss, which could 'over-filter' the solution of an inversion for emissions. Manual inspection combined with KNN statistical filtering was shown on two examples to be a promising way to confirm that outliers have a physically justified reason to be filtered for an atmospheric inversion that aims at quantifying the city emissions. However, the amount of data removed by this filtering approach is rather low and, therefore, the information from these data should not be statistically significant for the city scale inversions. We note however that it can be critical to discard them since the least square formulation of the optimization underlying these inversions could provide much weight to these data with large discrepancies to the model.

Within the city, the modeled $CO_2$ concentrations appear highly sensitive not only to the atmospheric vertical mixing close to the surface, but also to the prescribed temporal profile of anthropogenic emissions. These sources of errors are large, particularly in winter, and show a potential for biases that is problematic when aiming at the quantification of city emissions. Our results indicate that the temporal profile of the heating sector used by the AirParif inventory tends to bear a large uncertainty. It is one of the two major causes that led to the large model-data misfits during the nighttime. In the IdF region, $CO_2$ emissions from the heating sector are linked to the burning of gas and oil, and the electricity consumption. We could expect that a more constant diurnal profile should probably be a better approximation to the truth than the current one. This hypothesis has been further justified by an independent analysis of daily gas use and hourly electric consumption data within the IdF region (unpublished analysis led by a co-author of this study, François Marie Bréon). Furthermore, it remains difficult to interpret and use quantitatively in situ measurements within the city as long as there is no proper information about the turbulent airflow within and above the urban canopy. The near-surface mixing is not only controlled by the atmospheric stability conditions but also affected by the urban roughness and anthropogenic heat production. If the complex vertical mixing processes cannot be properly constrained in the transport model, it will be difficult to use the measurements acquired close to the sources in the atmospheric inversion system. Therefore, regular measurements of vertical $CO_2$ profiles, combined with relevant upper-air meteorological data (e.g., potential temperature and wind) and the mixing layer heights in the lower troposphere are expected to be included in the future Parisian $CO_2$ monitoring network. Such complementary measurements will be of great help to understand the characteristics of $CO_2$ vertical distribution under both stable and convective boundary-layer conditions. It can also be used to verify and validate the atmospheric transport model, and to reduce transport errors based on the data assimilation of more meteorological observations, leading to much higher accuracy in the atmospheric inversion system that aims at retrieving urban $CO_2$ fluxes.

In the suburbs, further away from the urban sources, the anthropogenic emissions are lower and the vertical gradient of $CO_2$ concentration, generated by the city emissions, is smoothed out by the atmospheric convection and diffusion processes. There is then less uncertainty than within the city about the efficiency of the vertical mixing. The link between the anthropogenic emission and the $CO_2$ concentration during the afternoon in winter can then be derived from the model with more confidence. However, the contribution of the biogenic flux to the $CO_2$ concentration is an issue during the growing season. The difficulty is mainly related to the simulation of the nocturnal $CO_2$ concentrations because of the large uncertainties in the atmospheric transport modeling as well as the biogenic fluxes. Focusing on the Paris region, two limitations of this study should be acknowledged and worth further investigating based on the high-resolution urban ecosystem modeling and monitoring so as to better quantify the impact of urban biogenic fluxes: (i) due to the coarse-resolution (1 km) SYNMAP land use data used for the VPRM model, the simulated biogenic fluxes in center Paris in this study are almost zero except for a few grid cells containing two big parks that are located in the eastern and western outskirts of the Paris city. While in reality, there are still a number of green space and pervious landscaped areas unevenly distributed in the city of Paris that need be considered with a fine-scale (sub-kilometer) model; (ii) there is a lack of

validation of the Paris-VPRM model in this study since no eddy covariance measurement is available within the Paris urban area and its surroundings. This limitation could be overcome by an expansion of the observation network with the neighborhood-scale urban eddy covariance flux measurements included. Moreover, additional measurements of carbon isotopes ($^{14}$C, $^{13}$C) and tracers coemitted with $CO_2$ (e.g., CO, $NO_x$) could be used to separate the contributions from fossil fuel and biogenic components to the total $CO_2$ concentrations, which would be beneficial for the optimization of sectoral $CO_2$ fluxes.

The influence of different $CO_2$ boundary conditions for our model domain is dependent on synoptic weather situations. As for the Paris region, the simulated $CO_2$ differences between CAMS and CarbonTracker are less than one ppm during most periods of westerly winds that bring in clean oceanic air masses, but they can vary by several ppm during some synoptic episodes, e.g. with north and easterly winds. This result advocates the practice of using additional observations to constrain the boundary inflow (e.g. Nickless et al., 2019; Mueller et al., 2018), or using $CO_2$ gradients when the wind direction is properly aligned with two (upwind-downwind) stations in the inversion of $CO_2$ fluxes of the Paris region (e.g. Bréon et al., 2015; Staufer et al., 2016).

## Author contribution

JL, FMB, GB and PC contributed to the design and implementation of the research. JL, FMB, GB, TL, BZ and PC contributed to the analysis and interpretation of the results. MR, IXR, SK and MH coordinated scientifically the development of the stations and ensured the production of the datasets. JL and FMB took the lead in writing the manuscript with input from all authors.

## Code/Data availability

All data sets and model results corresponding to this study are available upon request from the corresponding author.

## Competing interests

The authors have no competing interests to declare.

## Acknowledgements

This work is supported by the Ph.D. program funded by the IDEX Paris-Saclay, ANR-11-IDEX-0003-02 together with Harris Corporation. The instrumentation of the JUS site was funded by the National Research Agency ANR (CO2-MEGAPARIS ANR-09-BLAN-0222-03), Ville de Paris (Le CO2 Parisien project) and IPSL (MULTI-CO2 project). The COU equipment was funded by ANR (CO2-MEGAPARIS ANR-09-BLAN-0222-03) and EU KIC-CLIMAT (CarboCount City project). This latter also funded OVSQ and AND sites. We would like to thank Harris Corporation for the support of these ongoing analyses. Thanks also to Marc Jamous at Cité des Sciences et de l'Industrie (CDS), to the IPSL QUALAIR platform team and especially to Cristelle Cailteau-Fischbach (LATMOS/IPSL) for JUS, to OVSQ, and to LSCE/RAMCES technical staff for the maintenance of the in-situ $CO_2$ monitoring network, which was coordinated by Lola Brégonzio-Rozier (now at the Laboratoire National de métrologie et d'Essais, LNE Paris, France) and then further coordinated by Delphine Combaz.

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

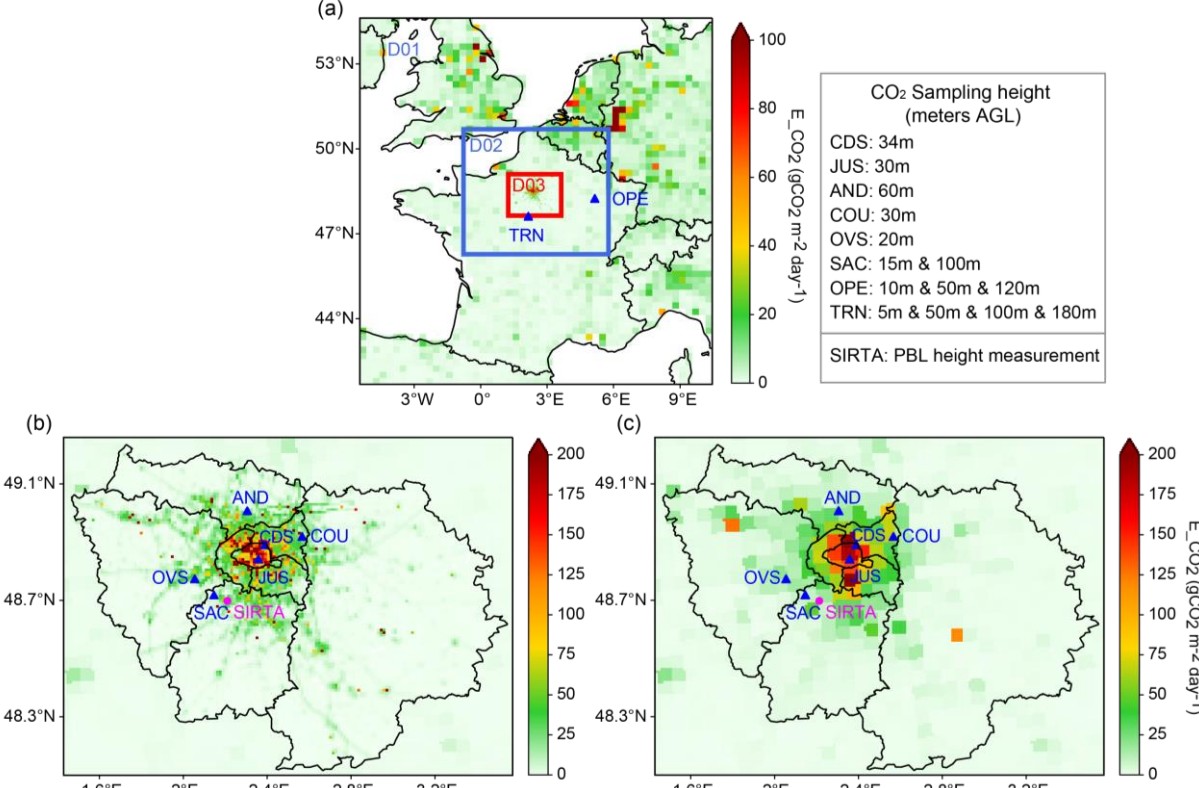

**Figure 1: (a) Three domains of WRF-Chem that are used for the simulations discussed in this study, together with the large-scale CO₂ emission for a weekday in November; Distributions of CO₂ emissions for a typical weekday in November from the (b) AirParif and (c) IER inventories. The bottom two maps show the location of six CO₂ measurement stations (blue inverted triangle), one PBL height measurement station (magenta circle) and the administrative limits of the Île-de-France region.**

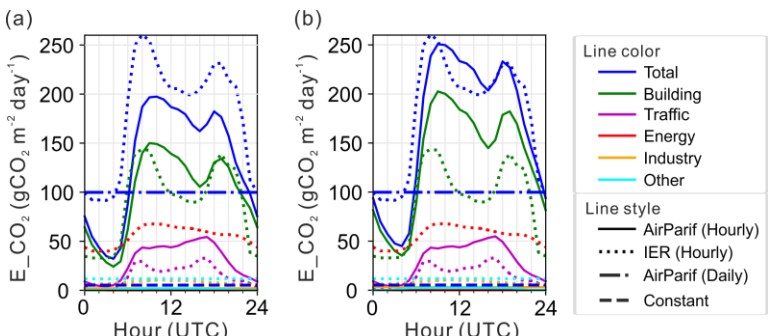

**Figure 2: Diurnal profiles (for January) of anthropogenic emissions in the grid cell of the emission map containing the CDS station. (a) is for the 1-km grid cell of the AirParif emission inventory that contains the station, whereas (b) is the 5-km grid cell of the IER inventory around the same station. The local time in Paris is one hour ahead of UTC (UTC+1) from November to March, and two hours ahead of UTC (UTC+2) from April to October.**

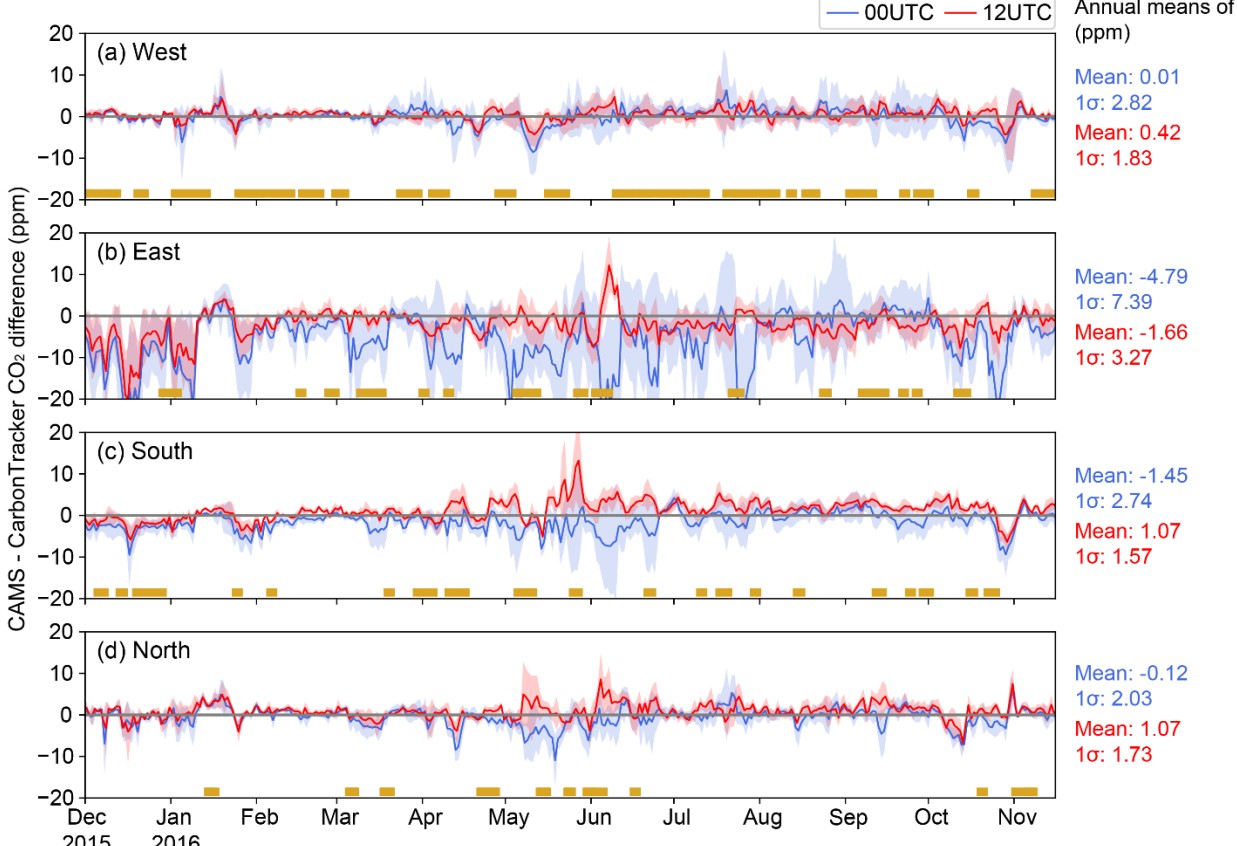

**Figure 3: Time series of average CO₂ concentration differences between CAMS and CarbonTracker at four lateral boundaries (west, east, south, north), averaged over the lowest 0.7 km AGL, of D01 for 00 UTC in blue and 12 UTC in red. The lines indicate the spatial means over each boundary (a latitudinal transect for western and eastern boundaries / a longitudinal transect for southern and northern boundaries). The shaded areas extend over one standard deviation (± 1σ) computed over the grid cells that make the lateral boundary (spatial standard deviation). The yellow symbols indicate the days when the wind blows from outside of the domain at the respective domain boundary. The numbers on the right side of the figure indicate annual means of (i) the spatial mean and (ii) the spatial standard deviation.**

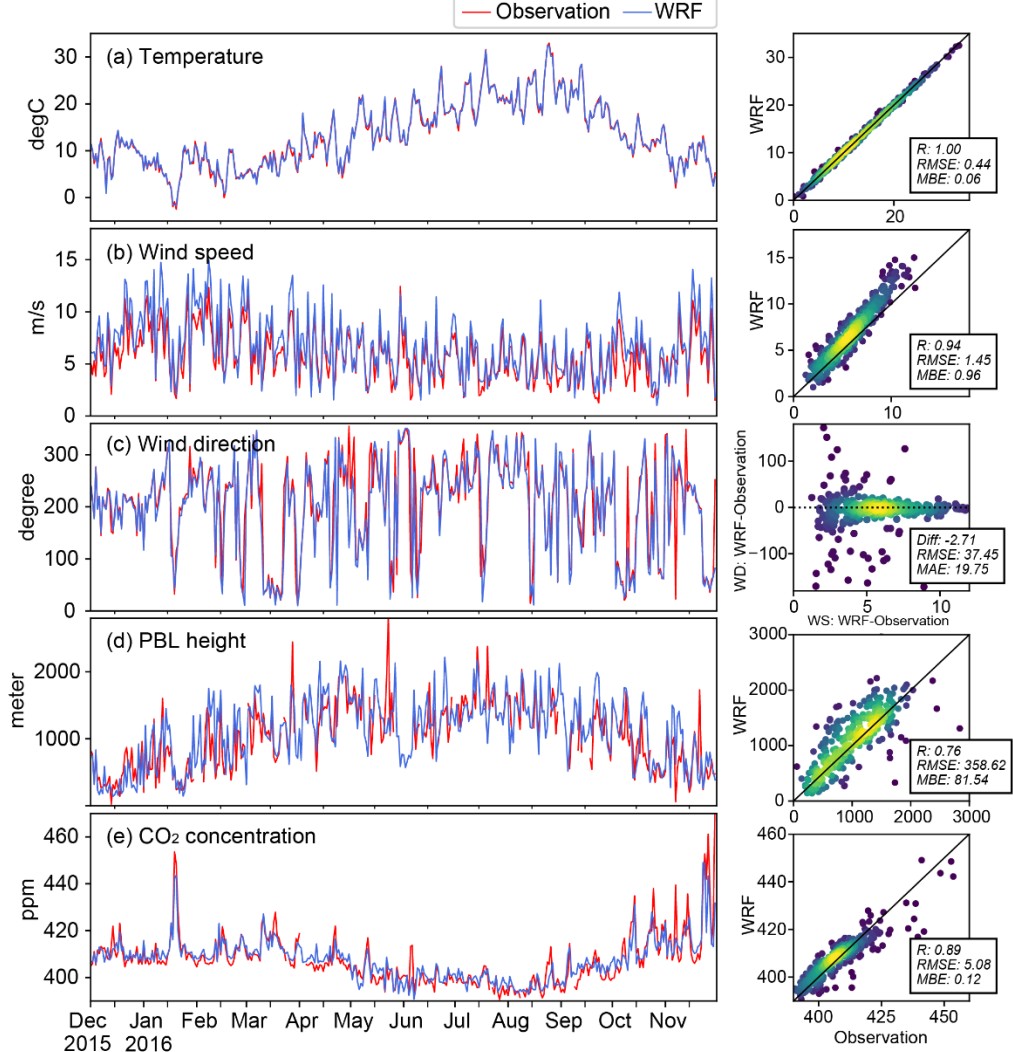

**Figure 4: Time series of the daily afternoon mean (11-16 UTC) observed and BEP_MYJ modeled (a) temperature, (b) wind speed, (c) wind direction and (e) CO2 concentration at SAC100 station. (d) Time series of the daily afternoon mean (11-16 UTC) observed and modeled PBL height at SIRTA station.**

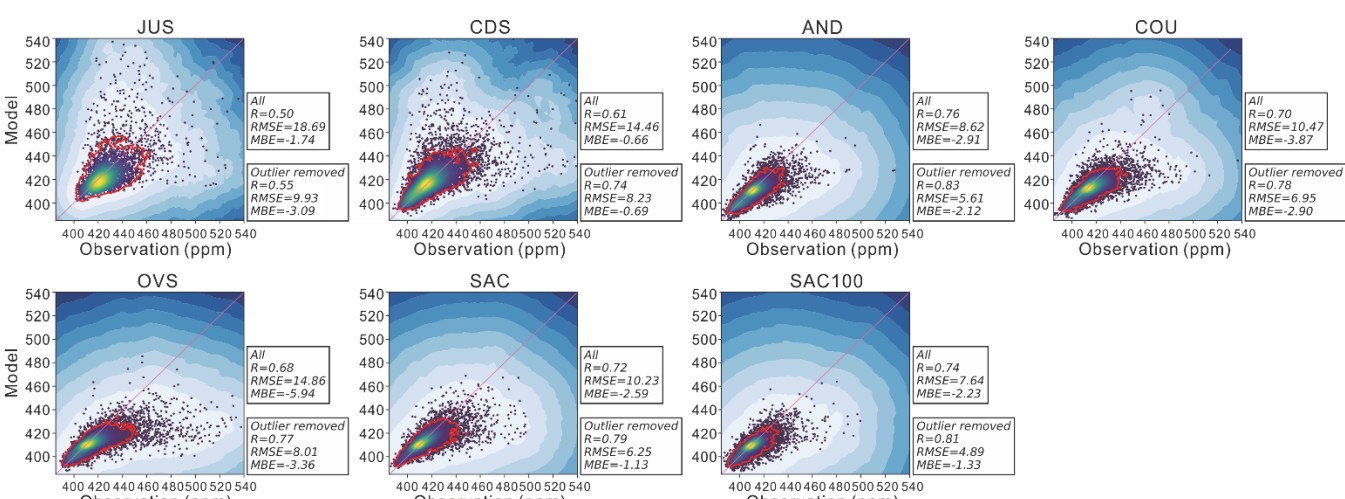

**Figure 5: Observed and BEP_MYJ (control run) simulated all hourly CO₂ concentrations at six monitoring sites from December 2015 to November 2016. The color of dots represents the density of points at a given position. The shade of blue area indicates the anomaly score for each point, with the minimum in dark blue and the threshold value in light blue. The dots lying outside the red contour (threshold value) are the large model-observation misfits (outliers) detected by the K-nearest neighbors (KNN) algorithm with an outlier fraction of 0.1.**

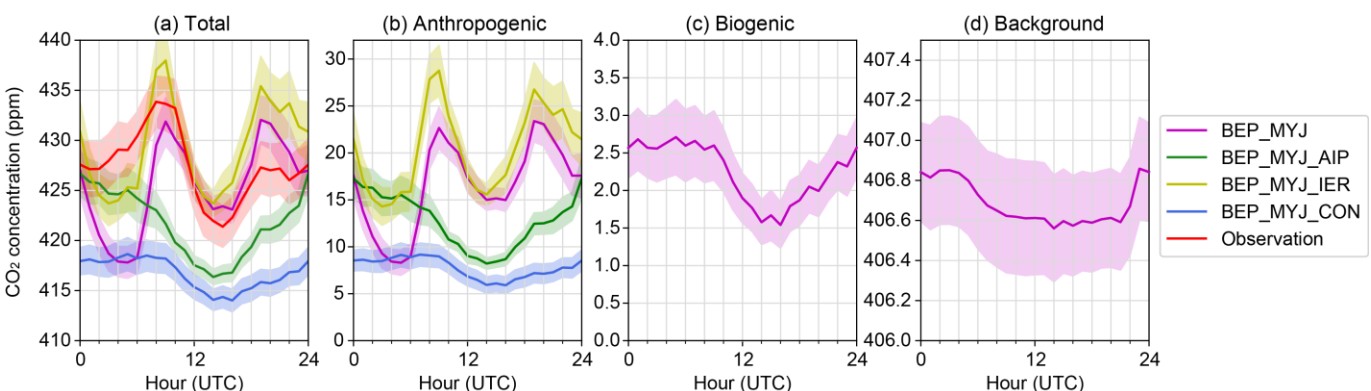

**Figure 6: Comparison of average diurnal variations between the (a) BEP_MYJ simulated (control run), (b) observed CO$_2$ concentrations, and (c) CO$_2$ differences between the model and the observations for four seasons. DJF denotes December-January-February, MAM denotes March-April-May, JJA denotes June-July-August and SON denotes September-October-November. The JUS instrument was not working during the summer of 2016.**

**Figure 7: Average diurnal cycle of (a) total, (b) anthropogenic, (c) biogenic and (d) background CO$_2$ concentrations in January for the four experiments at CDS station (Table 1b). The shaded areas extend over the standard deviation of the CO$_2$ concentration divided by the square root of the number of observations. BEP_MYJ is the control run using the AirParif inventory with hourly profile. BEP_MYJ_IER uses the IER inventory with hourly profile, BEP_MYJ_AIP uses a constant temporal profile (each pixel has a different emission, but constant in time based on the temporal average of the AirParif inventory). BEP_MYJ_CON uses a constant and spatially homogeneous emission where the emissions are distributed uniformly over the IdF whole territory.**

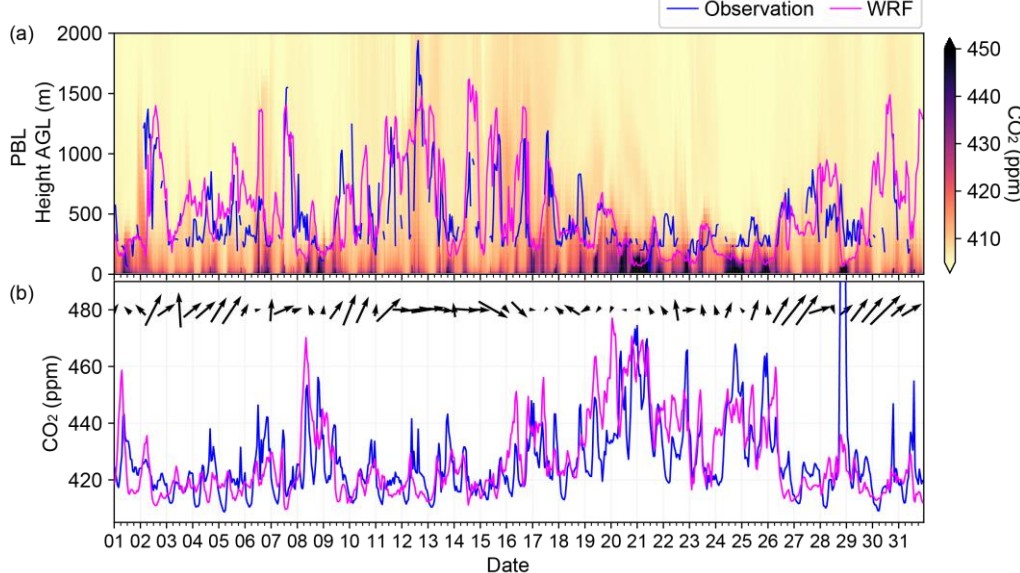

**Figure 8: (a) Vertical distribution of CO₂ concentrations at CDS station for January 2016, together with time series of the observed and BEP_MYJ simulated PBL heights at SIRTA station. (b) Time series of the observed and simulated CO₂ concentration at CDS. The arrows on the top of the figure indicate the wind speed and direction every day at noon and midnight. The simulation uses the AirParif CO₂ emission inventory.**

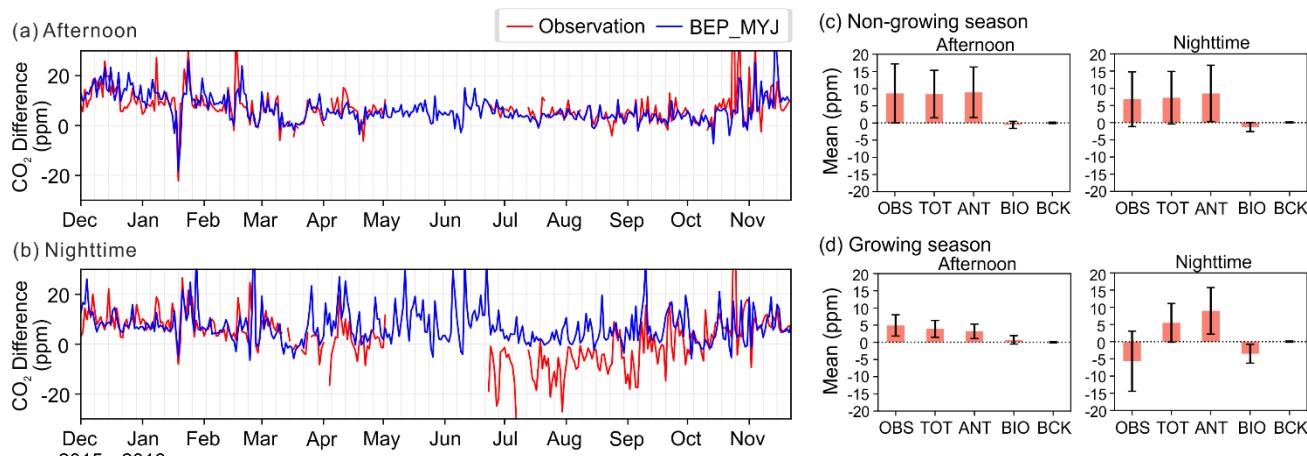

**Figure 9: Daily (a) afternoon mean (11-16 UTC) and (b) nighttime mean (21-05 UTC) CO₂ horizontal differences between CDS and SAC. Daily CO₂ horizontal differences between CDS and SAC from each sector for (c) Non-growing season from October to April and (d) Growing season from May to September. OBS indicates the observed CO₂ concentration differences. TOT, ANT, BIO and BCK indicates the simulated total, anthropogenic, biogenic and background CO₂ concentration differences respectively.**

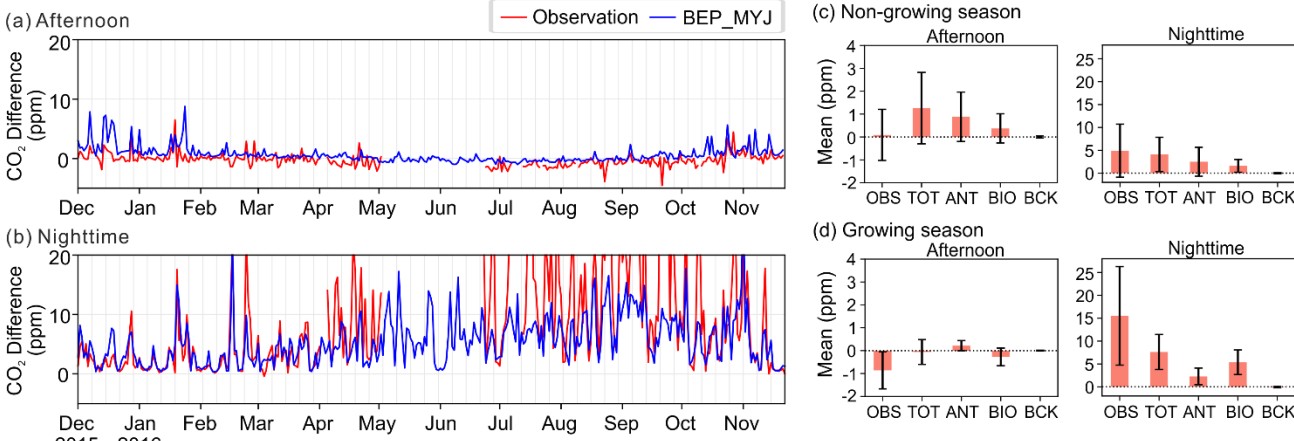

**Figure 10: Daily (a) afternoon mean (11-16 UTC) and (b) nighttime mean (21-05 UTC) CO₂ vertical differences at SAC (15m-100m).** Daily CO₂ vertical differences at SAC (15m-100m) from each sector for (c) Non-growing season from October to April and (d) Growing season from May to September. OBS indicates the observed CO₂ concentration differences. TOT, ANT, BIO and BCK indicates the simulated total, anthropogenic, biogenic and background CO₂ concentration differences respectively.

**Figure 11: Analysis of the "signal-to-noise" as discussed in the text for two periods of the day (afternoon 11-16 UTC, nighttime 00-05 UTC), and two months (January, July 2016). (a) is the median of the hourly standard deviation of the simulated near-surface CO₂ concentration computed among the five sensitivity runs (Table 1a); (c) is the same as (a) but for a vertical south-north slice that goes through the JUS station (shown as white dash lines); (b) is the median ratios of the hourly anthropogenic CO₂ concentration (average of the five sensitivity runs) to its respective standard deviation of the total CO₂ concentrations among the five sensitivity runs; (d) is the same as (b) but for the same vertical slice as in (c).**

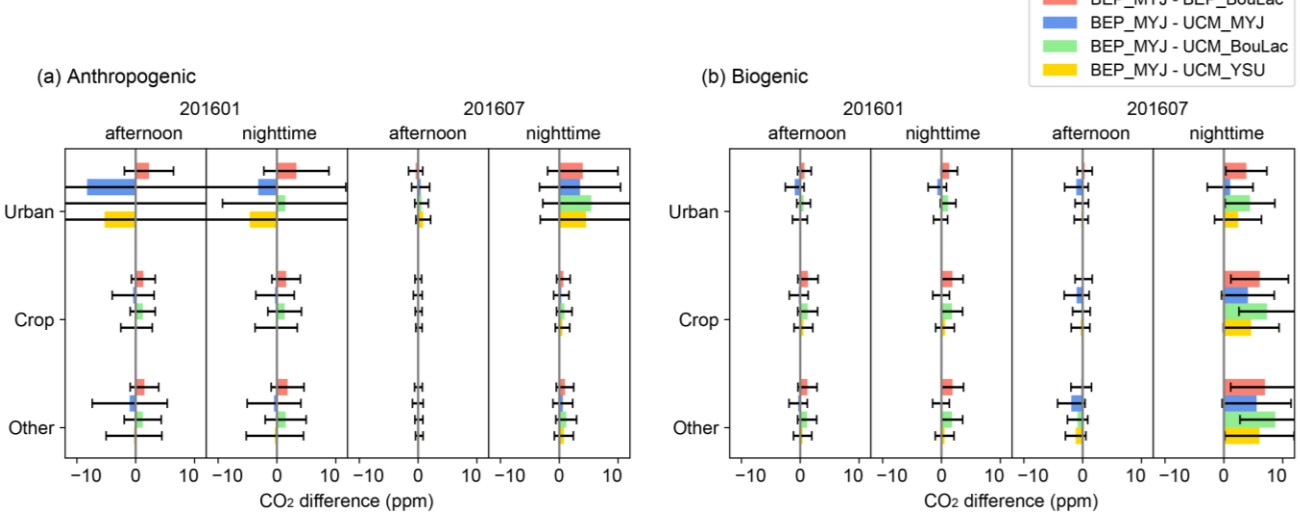

**Figure 12. Analysis of the $CO_2$ difference between the control run (BEP_MYJ) and each of the other four sensitivity runs over two one-month periods. The colored bars show the monthly mean difference whereas the black lines indicate +/- one standard deviation of the monthly values. The results are shown for two periods of the day (afternoon 11-16 UTC, nighttime 00-05 UTC) and for three land use types (urban, crop and the others).**

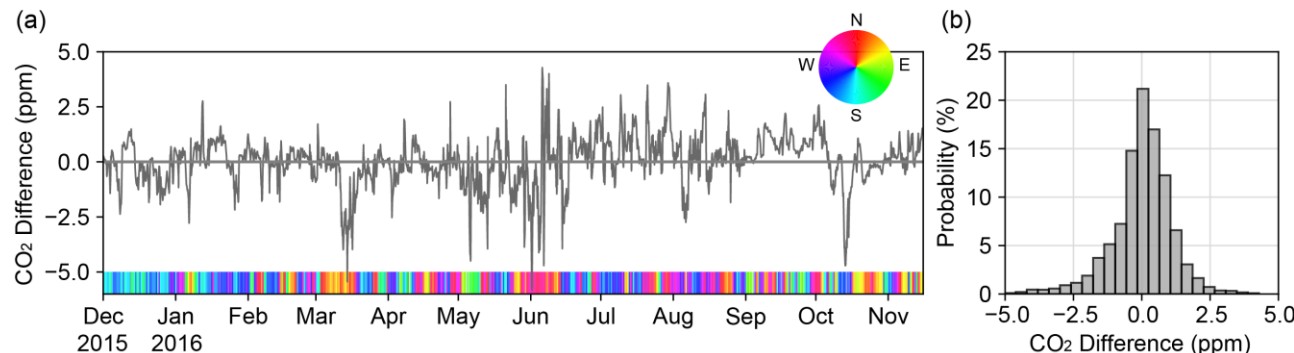

**Figure 13. (a) Time series and (b) Distribution of all hourly $CO_2$ concentration differences between BEP_MYJ and BEP_MYJ_CT using CAMS and CarbonTracker as $CO_2$ boundary conditions respectively. This comparison is based on the simulated values in the 25-km grid cell of the outermost domain (D01) containing the Paris city.**

**Table 1. Summary of WRF-Chem configurations used for the sensitivity experiments in this study. The bold text indicates the settings of the control run which is the same in all sets of sensitivity experiments.**

**(a) Sensitivity experiments of physics schemes carried out for a full-year period (2015.09-2016.11)**

| Configuration | PBL Scheme | Urban Canopy Scheme | Vertical Resolution | Anthropogenic Inventory | Boundary Condition |
|---|---|---|---|---|---|
| **BEP_MYJ** | MYJ | BEP | 44 levels (wherein 25 below 1.5 km). The lowest layer top is around 3.8 m AGL | IER (5 km, outside IdF) + AirParif (1 km, within IdF) with hourly profile | CAMS |
| BEP_BouLac | BouLac | | | | |
| UCM_MYJ | MYJ | UCM | 34 levels (wherein 15 below 1.5 km). The lowest layer top is around 19 m AGL | | |
| UCM_BouLac | BouLac | | | | |
| UCM_YSU | YSU | | | | |

**(b) Sensitivity experiments of anthropogenic emissions carried out for a one-month period (2016.01)**

| Configuration | Anthropogenic Inventory | | PBL Scheme + Urban Canopy Scheme | Boundary Condition |
|---|---|---|---|---|
| **BEP_MYJ** | IER (5 km, outside IdF) + AirParif (1 km, within IdF) | with hourly profile | MYJ + BEP | CAMS |
| BEP_MYJ_AIP | | without hourly profile | | |
| BEP_MYJ_IER | IER (5 km) | with hourly profile | | |
| BEP_MYJ_CON | Constant (5.28 $gCO_2/m^2/day$) | | | |

5   **(c) Sensitivity experiments of $CO_2$ boundary conditions carried out for the outermost domain (D01) for a full-year period (2015.09-2016.11)**

| Configuration | Boundary Condition | PBL Scheme + Urban Canopy Scheme | Anthropogenic Inventory |
|---|---|---|---|
| **BEP_MYJ** | CAMS | MYJ + BEP | IER (5 km) with hourly profile |
| BEP_MYJ_CT | CarbonTracker | | |