# Peer review of "Figure S1: Time series of average CO2 concentration differences between CAMS and CarbonTracker at four lateral boundaries (west, east, south, north), averaged over vertical layers above 0.7 km AGL, of D01 for 00 UTC in blue and 12 UTC in red. The lines indicate the spatial means over each"

_Atmospheric Chemistry and Physics, 2020_

## Referee Comment (RC1) · Anonymous Referee #1 · 31 Jul 2020

General Comments

The study investigates potential sources of error for estimating urban CO2 emissions using atmospheric observations. Understanding and mitigating theses errors is necessary to produce more accurate emission estimates, and the authors suggest several criteria to select data to avoid the impact of these error sources. While the study focuses on Paris, the methods and results presented are more widely applicable and of interest to other urban emission estimation schemes using atmospheric data and transport models.

The methods of the paper focus on examining a set of clearly described WRF-

[Figure]

Chem forward model runs, with the $CO_2$ emissions, boundary conditions and physics schemes varied. These comprise a logical set of factors to explore, with the authors acknowledging this is a subset of all possible error sources but is still shown to be important. The detailed analysis of these results links well with the corresponding conclusions drawn (suggestions ii and iii of the abstract). I believe this paper is a useful contribution to the field, providing quantifications of significant sources of uncertainty in current models and providing a framework for further urban systems to examine their own uncertainties. However, I do have a few concerns with other aspects of the paper, as detailed below. Therefore, I recommend this paper for publication in ACP once the issues outlined below have been addressed.

Specific Comments

Introduction - The study could be seen as an extension to previous works in looking at sources of uncertainty (Martin et al 2019 https://doi.org/10.1016/j.atmosenv.2018.11.013) and are complementary to other recent studies on uncertainties in estimating urban emissions (such as Balashov et al 2020 https://doi.org/10.5194/acp-20-4545-2020). The context set out in the paper could be improved by including comparisons to such other studies.

Page 1 line 28 – Value quoted is for scope 2 emissions, but inversions only estimate scope 1 emissions. Either this should be made clear or the authors should use scope 1 emissions value.

Page 6 line 6 – The use of the KNN outlier removal needs greater justification and is my greatest concern with this paper. The authors claim that this algorithm removes observations of sources too local to be resolved or meteorological conditions that model is less skilled with (which are valid reasons for removing data points) but provides no evidence that this is the case. As it is, the algorithm may just be arbitrarily throwing away data that highlights systematic over or underestimates in the emissions field that is needed for an inversion. Either it needs to be demonstrated that the algorithm only

removes points that are linked to these conditions, or a different method, preferably based on physical reasoning, should be used.

Page 7 line 2 – The authors say that individual measurement errors are negligible compared to model-data differences. However, model-data comparisons are made on hourly time scales and there will be (potentially large) variation within the hour. How is this sub-hour variation accounted for in model-data comparison?

Page 8 line 17 – The authors suggest upwind-downwind gradient can be used even in growing seasons as natural biogenic fluxes do not completely offset anthropogenic fluxes – but this still requires good knowledge of biogenic fluxes as they will make a major contribution to the observed mole fraction difference. The authors note several lines later that the biogenic fluxes show systematic errors, does this contradict the first statement?

Page 10 lines 1-21 (and Figure 11) – This section should be reworked for clarity. My understanding is that the authors have averaged the difference of tracer concentrations between runs across time (by month) and horizontal space (by land type) to calculate the individual values shows in figure 11 – but this should be made clearer in the writing. For many of the values, the standard deviation is large w.r.t. the mean difference. A different type of plot that shows the distribution, such as a violin plot, may be more appropriate.

Page 11 line 13 – A strong conclusion for the use of KNN outlier removal that is not justified, see above.

Figure 12 – Why use a cumulative distribution and not a histogram – what are the authors trying to show with this choice?

Technical Comments

Page 8 line 18 – suggest "(the SAC station had unfortunately measurement gaps)" -> "(unfortunately the SAC station had measurement gaps)" and dates of gap added

General note on figures – Rainbow colour schemes should be replaced with perceptually uniform colour schemes, and red-green colour schemes should be avoided due to Colour vision deficiency (colour blindness)

Figure 4 – Both the dark blue filled contour and red contour are called the 'threshold' but the two are not in agreement (red contour seems to be the correct one)

Figure 6 – A note to remind the reader that this figure is for January only would be helpful

Figure 7 – This figure is dense, which hinders clarity. I suggest the wind direction and mf timeseries to moved to new windows with a shared x axis. The boundary layer height should also have a larger contrast to make it more visible

Figure 10 – Showing the line of transect for the south-north slice on the lat-lon plots would make interpretation of the figure clearer

---

## Referee Comment (RC2) · Anonymous Referee #2 · 3 Aug 2020

**Review of Lian et al., "Quantitative evaluation of the uncertainty sources for the modeling of atmospheric CO2 concentration within and in the vicinity of Paris city"**

*General comments:* The authors attempted to determine the uncertainty sources for the modeled CO2 concentrations over Paris, France, using a set of WRF-Chem simulations varying with physics-based transport, fossil fuel emissions, and CO2 boundary conditions, for 2016. They mainly focused on the impact of PBL schemes and with the combination of the urban canopy models, two fossil fuel emission inventories with /without hourly variability, and two global models as boundary conditions on the modeled CO2 in comparison with the ground-based in-situ CO2 measurements. Their results show that model-data mismatch maximizes in the nighttime so they recommended the readers to discard the model-data misfits and use afternoon measurements for inversion. This is not new, and I believe that is what we do in atmospheric inversion. They also found the boundary condition could cause large differences at the synoptic scale and suggest using additional observation to constrain boundary conditions. This is also not new. The authors are aware of these points because they cited those papers. So, I failed to locate the novelty of the work that brought into the community. The authors, in my opinion, have repeated some of the previous studies without extending the science further.

Besides, the authors cited a few pilot CO2 urban studies, such as INFLUX and LA megacity. Both Lauvaux (2016) and Feng (2016) pointed out the significant improvement of using highresolution fossil fuel inventories in simulating CO2 at the urban environment. Although the authors included two different fossil fuel inventories in the simulation with the variation of the temporal components, I have a hard time following the goal of the experimental design. I thought they would explore the sensitivity of modeled CO2 to the temporal resolutions when I was reading the methods and section 3.2.1, but the related findings were not emphasized in the conclusion. Why?

One of the major concerns in the urban CO2 studies falls in the impact of the biosphere around or within the cities. The results of this study also showed that the impact of the biogenic fluxes is significant and not negligible, meaning that the biosphere is another uncertainty source over Paris. The author can refer to Feng (2019a; 2019b) to construct a set of biospheric fluxes and investigate the uncertainty of the biospheric in the modeled CO2 as well. Additionally, the authors relied on the VPRM module in WRF-Chem to provide the biogenic fluxes. It's not clear to me that if VPRM has been tuned with flux towers or not. The VPRM parameters in WRF-Chem are fixed and needed to be tuned with the flux towers to have a relatively accurate biospheric flux estimation (Hilton et al., 2013: Hilton et al., 2014). If the authors used the default values for the VPRM parameters in this study, the authors will have to consider the errors caused by the biosphere during the interpretation of the results which is almost impossible to isolate from transport, emission, or boundary condition. However, because of the simplicity of VPRM, the authors can build a set of parameter-based perturbations of the biospheric fluxes via VPRM to address my first concern.

The authors chose five combinations of the PBL schemes and urban canopy models to study the impact of the transport and concluded that it's difficult to have "good" transport. First of all, what are the rationales the authors believe these tow schemes are the key players of the CO2

urban modeling? Díaz-Isaac (2018) using an ensemble approach pointed out PBL indeed is a major player but ranked No. 2. The most dominant parameterization is the land surface model used in the study. I am aware that the response of the modeled CO2 to the model physics may vary when the location changes. Have the authors explored the impact of other model parameterizations on the simulated CO2? This may be also why the model results have such a large bias in this study. Secondly, the model-data mismatches are extremely large and out of my expiation. For example, the whole year averaged diurnal mismatch can be as large as -10 to 5 ppm at the two urban sites in Figure 5. I found a similar figure in Figure 9 of Feng (2016), even though it's a month averaged value, in which the diurnal cycle from the high-resolution simulation looks almost identical to the observation. What causes the large bias in this work? I would check if any errors caused by other model physics. Thirdly, the authors concluded that the transport issue is difficult to identify. I disagree. The model transport can be evaluated with meteorological observations. Apparently, there are meteorological observations at the monitoring sites. Additionally, there are quite a few WMO stations in the domain. Comparing with meteorological observations in the model domain will allow the authors to have a better sense of the model transport.

As the authors mentioned that boundary conditions can lead to large bias in the inversed results, the results showed that 5-20 ppm day-to-day difference between the two global models along the edges of the model domain. In the CO2 regional (inverse) modeling, one of the major concerns about the boundary condition is the conservation of mass (Butler et al., 2020). How did the authors handle mass conservation when incorporating global modeled CO2 into the regional model domain? Another issue is that the number of boundary conditions used is too small to quantify the uncertainty. Strictly speaking, to be able to claim quantification of the uncertainty sources, a large number of the ensemble and a set of calibration procedures are required, such as rank histogram, reliability diagram, brier scores, etc. (Garaud and Mallet, 2011). Although it may be difficult to meet two criteria with the CO2 modeling, the authors will at least need three of them to study the sensitivity.

In summary, this work claims that it has a quantitative evaluation of uncertainty sources in the CO2 modeling, but the experimental design is far from achieving the goal. It eventually is merely a sensitive study of modeled CO2 to the selected fossil fuel emissions, the combination of PBL and urban canopy models, and boundary condition. The size of the ensemble they built does not allow them to do a solid quantification study. As I mentioned, this study appears repeating some of the previous studies without advancing the understanding the community already holds currently, neither in science nor in techniques.

There are no clear rationales why they made such selection as I pointed out with the transport "ensemble". The authors did not address the major issues in urban modeling, i.e., the impact of biosphere, and regional modeling, i.e., the conservation of mass when applying boundary conditions. They also failed to have a clear conclusion about the findings associated with fossil fuel emissions. In my opinion, this work is incomplete and must be extended to consider publication; these concerns I brought up can be addressed, which, however, will require a new design of the method. In addition to the specific comments I listed below, I would not recommend this MS to be a published in ACP.

**Specific comments:**

Section 2: There are important details missing in the description of the model setup.

- 1) Did the model use simulation cycles? If yes, how often is it? If yes, how was the CO2 field addressed, initializing every time or being carried over simulation cycles?
- 2) ERA-Intrim and the outermost domain of WRF-Chem have quite different resolutions. What are the rationales that the authors used grid nudging over spectral nudging?
- 3) As I mentioned in the general comments, has the VPRM parameters constrained with the flux tower measurements?
- 4) When the authors were incorporating CO2 IC/BC to WRF-Chem, how did the author address the conservation of CO2 mass?
- 5) When using global modeled CO2 as IC, the discontinuity of the global and regional model dynamic can cause discrepancy of the CO2 as well. How much the difference caused by the discontinuity would be?

P 6, L 25-30: the author interoperated that the reason of the higher CO2 concentrations in Fall than in winter was due to the anticyclone keeping the high CO2 in the domain for quite a while. I disagree. If it's due to meteorology, the impact on the fossil fuel CO2 and biospheric CO2 concentration should be the same. We should see lower CO2 in the suburban sites, but we don't.

P7, L1-5: as I said earlier, the authors should be able to identify at least to some degree if the issues are in transport or boundary conditions by comparing with the meteo data.

P10, L10-15: what causes the different bias between the BEP and UCM schemes? I would like to see a deeper explanation of that instead of simply saying lower or higher.

P11, L19-21: I agree that based on the current setup, there is little hope to improve the model performance. However, the authors can follow my suggestion listed in the general comments. For example, checking the land surface model used, comparing with meteo data, etc., to identify if the problem is caused by transport is the first step. Then the authors can look into the emission, boundary conditions, etc.

Figure 5: please use local time in the x-axis instead of UTC. The much bigger issue is the large bias in the biases.

**Reference:**

Butler, Martha P., Thomas Lauvaux, Sha Feng, Junjie Liu, Kevin W. Bowman, and Kenneth J. Davis. "Atmospheric Simulations of Total Column CO2 Mole Fractions from Global to Mesoscale within the Carbon Monitoring System Flux Inversion Framework." Atmosphere 11, no. 8 (August 2020): 787. https://doi.org/10.3390/atmos11080787.

Díaz-Isaac, Liza I., Thomas Lauvaux, and Kenneth J. Davis. "Impact of Physical Parameterizations and Initial Conditions on Simulated Atmospheric Transport and CO2 Mole Fractions in the US Midwest." Atmospheric Chemistry and Physics 18, no. 20 (October 16, 2018): 14813–35. https://doi.org/10.5194/acp-18-14813-2018.

Feng, S., Lauvaux, T., Newman, S., Rao, P., Ahmadov, R., Deng, A., et al. (2016). Los Angeles megacity: a high-resolution land–atmosphere modelling system for urban CO2 emissions. Atmospheric Chemistry and Physics, 16(14), 9019–9045. https://doi.org/10.5194/acp-16-9019-2016

Feng, Sha, Thomas Lauvaux, Kenneth J. Davis, Klaus Keller, Yu Zhou, Christopher Williams, Andrew E. Schuh, Junjie Liu, and Ian Baker. "Seasonal Characteristics of Model Uncertainties From Biogenic Fluxes, Transport, and Large-Scale Boundary Inflow in Atmospheric CO2 Simulations Over North America." Journal of Geophysical Research: Atmospheres 124, no. 24 (2019): 14325–46. https://doi.org/10.1029/2019JD031165.

Feng, Sha, Thomas Lauvaux, Klaus Keller, Kenneth J. Davis, Peter Rayner, Tomohiro Oda, and Kevin R. Gurney. "A Road Map for Improving the Treatment of Uncertainties in High-Resolution Regional Carbon Flux Inverse Estimates." Geophysical Research Letters 46, no. 22 (2019): 13461–69. https://doi.org/10.1029/2019GL082987.

Garaud, D., and V. Mallet. "Automatic Calibration of an Ensemble for Uncertainty Estimation and Probabilistic Forecast: Application to Air Quality." Journal of Geophysical Research: Atmospheres 116, no. D19 (October 16, 2011). https://doi.org/10.1029/2011JD015780.

Hilton, T.W., K. J. Davis, and K. Keller. 2014. Evaluating terrestrial CO2 flux diagnoses and uncertainties from a simple land surface model and its residuals, Biogeosciences, 11, 217-235, doi:10.5194/bg-11-217-2014.

Hilton, T.W., K. J. Davis, K. Keller, and N.M. Urban. 2013. Improving North American terrestrial CO2 flux diagnosis using spatial structure in land surface model residuals, Biogeosciences, 10,4607–4625, doi:10.5194/bg-10-4607-2013.

---

## Referee Comment (RC3) · Anonymous Referee #3 · 7 Aug 2020

General Comments: This study by Lian et al., 2020 attempts to identify and quantify significant sources of errors that can hinder the accurate estimation of urban-scale emissions. The objective of this study, as claimed by the authors, also includes demonstrating how these diagnostics can be used for inverse modelling studies. An ensemble of WRF-Chem simulations are performed, varying emission inventories (one month of simulations), PBL schemes and urban canopy schemes (one year of simulations), and boundary conditions (one year of simulations). The topic is fascinating and is essential to investigate the how sensitive is the emission estimate to the different components of the transport mechanisms (simulated by the model), flux variations, and assumptions/methods employed. This is a well-written manuscript with clearly described methods and results arranged in a logical order, which made the manuscript easy to follow. The conclusions drawn, based on their analyses, are reasonable and are applicable to those working on city-scale and mesoscale inversions of CO2 using WRF-Chem.

The study in this present form, however, fails to justify the title. Though the study considered an ensemble of simulations and subsequent analyses, it is still insufficient to make a quantitative estimation/evaluation of sources of errors in CO2 simulations which is adequate to the broad spectrum of inverse modelling studies. I'd instead consider it as a study on diagnosing the effect of vertical mixing (PBL schemes), specific modelling criteria (urban schemes) as well as boundary conditions in city-scale modelling, in addition to assessing the sensitivity of simulations to the emission patterns. I believe that the title can be reworked accordingly to present the study appropriately. Additionally, some other sections require more clarification, modification, and further analysis, as mentioned below. Thus, I would recommend a major revision before considering for publication in ACP.

Though it is mentioned as one of the objectives, I don't see that the study has addressed the question to what extent or how the model-measurement error can be reduced. This is a major concern of mine. I'd consider that the authors could devise efficient analysis strategies to address this, given the availability of measurements from 6 +2 sites and ensemble of simulations. An adequate diagnosis of model-measurement mismatches is missing, which is a weakness of this manuscript. A discussion on how the diagnostic results can be used for the betterment of inversion studies is vaguely articulated in the manuscript. All the guidelines for the data selection put forward by the study (such as discard data with high model-data misfits, use only afternoon values; test the influence of boundary conditions) is already known to the community and currently practised in inverse model calculations. I would suggest authors avoid the above statement of objective or revise thoroughly while incorporating additional analysis to explain the novelty of their findings.

Specific Comments: Fig. 2. For Line style descriptions, please use another colour (e.g.

black) than those used as line colours. Blue is already used for "Total". I had a hard time to understand. Also, I'd suggest you remove (c) and (d) and include AirParif (daily) and Constant in (a) and (b).

Differences between CAMS and CarbonTracker at the four lateral boundaries: Why are there substantial differences between daytime and nighttime in East boundaries, sometimes even in opposite phases (Fig. 3b)? Please explain.

Evaluation with in situ observations: I am not very convinced with the usage of KNN method? How is the outlier fraction calculated? How sensitive is the filter size in another outlier fraction? What are the criteria for the choice of 0.1 in this case? Given that outliers are removed, why the model-observation mismatch is this high (Fig. 4)? How can these mismatches be reduced? How about background sites in terms of evaluation? In addition to reporting the mismatches, I'd suggest the authors explain the reasons for these large deviations from observations. This is critical as I also see unexpectedly significant model-measurement differences in diurnal averages. Have authors checked different choice of physics/dynamics schemes or other parameters available in WRF-Chem to reduce this mismatch?

Sect. 3.2.1 "modeled value (green line in Figure 6) gets somewhat closer to the observation" In Fig. 6, the blue curve represents constant emissions. Please check. Also please indicate the season (or January) in the figure caption. The simulations (BEP_MYJ_CON) reproduce the observed patterns better than other simulations; however, not in magnitude. So please rephrase the sentence accordingly: "modeled value (green line in Figure 6) gets somewhat closer to the observation". Also, I am happy to note that the authors demonstrate the effect of emission trend and atmospheric vertical mixing here. Please comment on the effect of boundary conditions (though I expect it to be minimal by looking at the patterns in BEP_MYJ_CON).

PBLH and vertical distribution of the modelled CO2 (BEP_MYJ): It is not clear to me why authors have mentioned Nielsen-Gammon et al., 2008 for PBLH estimation. What

extent the MYJ scheme and Nielsen-Gammon et al., 2008 differ in deriving the PBLH? If different, a comparison plot will be helpful here. I am a bit surprised with the high PBL values in winter (initial half-month) over Paris. Do authors look at the PBL measurements (e.g. lidar measurements)? Fig. 7 is confusing as the 34-m AGL curves have nothing to do with the left Y-axis values. I would suggest authors separate these two curves (magenta and pink) from this and make an independent plot along with PBLH.

Sect. 4: Please see my comments above (w.r.t title) and refine this section thoroughly.

Minor comments: Page 7, "from 18 pm to 22 pm)" Please change to 18:00 UTC to 22:00 UTC.

---

## Author Comment (AC1) · 29 Apr 2021

We would like to thank Referee #1 for his/her thoughtful comments and detailed suggestions to our manuscript. In the following, we answer to the reviewer's comments and indicate the changes in the manuscript that were implemented according to the recommendations. The comments are in black. Our answers are in blue. All the figure numbers correspond to the revised manuscript.

**Anonymous Referee #1**

**General Comments**

The study investigates potential sources of error for estimating urban $CO_2$ emissions using atmospheric observations. Understanding and mitigating these errors is necessary to produce more accurate emission estimates, and the authors suggest several criteria to select data to avoid the impact of these error sources. While the study focuses on Paris, the methods and results presented are more widely applicable and of interest to other urban emission estimation schemes using atmospheric data and transport models.

The methods of the paper focus on examining a set of clearly described WRF-Chem forward model runs, with the $CO_2$ emissions, boundary conditions and physics schemes varied. These comprise a logical set of factors to explore, with the authors acknowledging this is a subset of all possible error sources but is still shown to be important. The detailed analysis of these results links well with the corresponding conclusions drawn (suggestions ii and iii of the abstract). I believe this paper is a useful contribution to the field, providing quantifications of significant sources of uncertainty in current models and providing a framework for further urban systems to examine their own uncertainties. However, I do have a few concerns with other aspects of the paper, as detailed below. Therefore, I recommend this paper for publication in ACP once the issues outlined below have been addressed.

We thank the reviewer for these very supportive comments.

**Specific Comments**

Introduction - The study could be seen as an extension to previous works in looking at sources of uncertainty (Martin et al 2019 https://doi.org/10.1016/j.atmosenv.2018.11.013) and are complementary to other recent studies on uncertainties in estimating urban emissions (such as Balashov et al 2020 https://doi.org/10.5194/acp-20-4545-2020). The context set out in the paper could be improved by including comparisons to such other studies.

Please see our answer to Referee #2. Table R1 shows a comparison between this study and few pilot $CO_2$ urban studies with the objective to investigate in detail the sources of uncertainty/error in the atmospheric $CO_2$ modeling for cities, such as Los Angeles (Feng et al., 2016) and Washington DC/Baltimore (Martin et al. 2019). Given that the sources and characteristics of urban fluxes of $CH_4$ is different from those of $CO_2$, we have not included a comparison with the Balashov et al. 2020 here.

Page 1 line 28 – Value quoted is for scope 2 emissions, but inversions only estimate scope 1 emissions. Either this should be made clear or the authors should use scope 1 emissions value.

As suggested, we have used the value of direct emissions (scope 1). The modified text is as follows:

"cities directly release about 44 % of the global energy-related $CO_2$ emissions"

Page 6 line 6 – The use of the KNN outlier removal needs greater justification and is my greatest concern with this paper. The authors claim that this algorithm removes observations of sources too local to be

resolved or meteorological conditions that model is less skilled with (which are valid reasons for removing data points) but provides no evidence that this is the case. As it is, the algorithm may just be arbitrarily throwing away data that highlights systematic over or underestimates in the emissions field that is needed for an inversion. Either it needs to be demonstrated that the algorithm only removes points that are linked to these conditions, or a different method, preferably based on physical reasoning, should be used.

In response to the reviewer's concerns about the KNN outlier removal, we did perform some further analyses and validations of this method to support the approach and the related statements in the manuscript. These analyses definitely show that the outliers generally correspond to either:

1) the model's inability under specific meteorological conditions.

After analyzing the dates of the identified outliers, we found clusters of outliers that occur as the result of weather episodes with a duration of one-to-few days. Several cases were identified and described in Table S1. One sample case, presented here, shows unfavorable meteorological conditions from Jan 18th to 21st 2016. During this 4-day period, with a return of the winter anticyclonic conditions over the entire region, dense fog and weak winds were observed. Stubborn low clouds kept temperatures chilly with little snow. Figure S3a shows the time series of the observed and modeled (using MYJ_BEP) hourly $CO_2$ concentration at SAC station. The grey shaded areas indicate the ranges of model results with five physical parameterization schemes used in this study (Table 1a in the manuscript). The yellow vertical lines indicate the large model-observation misfits (outliers) detected by the KNN algorithm. It can be seen that for the certain hours that were tagged as outliers, the differences between observed and modeled $CO_2$ concentrations can be as large as 70 ppm. Meanwhile, the spread of the simulations of $CO_2$ is much larger than during the days before and after this period, leading to a higher mean bias error and root-mean square error of the ensemble mean. Figure S3b shows the distribution of the hourly $CO_2$ concentrations as a function of the wind speed for the year 2016. It clearly illustrates that the detected outliers occurred more often in weak-wind conditions ($< 2.5$ m/s) which are difficult to reproduce by the model. From this example, we can say that KNN can detect outliers corresponding to conditions when the model physics encounters limitations.

2) the specific measurement contaminations from local unresolved sources of $CO_2$ emissions.

On the other hand, this KNN method was inspected for its ability to remove some $CO_2$ spikes due to very local influences or sampling contaminations, mainly under low wind speed conditions. We illustrate this phenomenon with the example of the measurements of hourly CO and $CO_2$ concentrations (CO being used to confirm the anthropogenic origin of the spikes in the atmospheric concentration) at the OVS station in 2016. The CO and $CO_2$ hourly mole fractions, as well as their ratios, are plotted as a function of observed wind speed and direction (Figure S4a). The location of the CRDS CO & $CO_2$ sampling inlet is on a building roof, where there is a building ventilation exhaust shown in Figure S4b. Figure S4a shows that the CO signal tends to be larger relative to that of $CO_2$ with low winds ($< 4$ m/s) blowing from the east. This corresponds to the position of the building exhaust air system relative to that of the sampling inlet, and this is at odd with the North East position of the Paris urban area or of the main neighbor and large area sources relative to the OVS site. Further investigation shows that these CO spikes at OVS are mostly measured at night in winter, leading to a nighttime mean concentration even much larger than those two urban stations (JUS and CDS). We thus highly suspect that the measurements of CO and $CO_2$ are contaminated by the exhaust air of the building under specific conditions (winter nighttime with light winds). Most of the dates corresponding to these CO and $CO_2$ spikes exactly coincide with the outliers at OVS that have been detected by the KNN algorithm shown in Figure 5 in the manuscript. From this example, we can say that KNN can detect outliers (in the data) corresponding to real physical local contaminations.

Therefore, the KNN method, as shown above, can detect misfits between the observations and the models that would be misleading for the city scale inversions. But we also acknowledge the fact that removing data points simply based on statistical analysis without identifying the outliers on a case-by-case basis may lead to a loss of data that are suitable for the city scale inversion. In practice, manual inspection is preferable for the identification of the cause of the error. However, this is not practical given a large amount of data at six in situ stations collected over one year as those analyzed in this study. It is also difficult to find a general outlier detection method fitting to any site, model and atmospheric transport conditions.

The above analyses, table and figures will be put into the supplement. We have added the following text in section 3.1.2:

"We further analyzed the filtered hourly concentrations (detailed in supplement material Figure S3 and S4) and confirmed the contamination at one of our sites (OVS) and the relationship between meteorological conditions and excluded modeled concentrations."

Regarding the conclusion and discussion section, we agree with the reviewer that it would be more appropriate to encourage the use of the KNN algorithm based on a deeper analysis of the detected outliers instead of just saying one should use it as crude data filtering. We have rephrased the text as follows:

"We should also note that removing outliers based on statistical analysis without attributing them to a real data contamination or model limitation has potential for data loss, which could 'over-filter' the solution of an inversion for emissions. Manual inspection combined with KNN statistical filtering was shown on two examples to be a promising way to confirm that outliers have a physically justified reason to be filtered for an atmospheric inversion that aims at quantifying the city emissions. However, the amount of data removed by this filtering approach is rather low and, therefore, the information from these data should not be statistically significant for the city scale inversions. We note however that it can be critical to discard them since the least square formulation of the optimization underlying these inversions could provide much weight to these data with large discrepancies to the model."

Table S1. Meteorological conditions for several situations when large model-data misfits have been detected by the KNN algorithm.

| Date | Bulletin Climatique Météo-France* |
|---|---|
| January 19-21 2016 | With the anticyclonic conditions, frosty fogs and stubborn low clouds were observed. Temperatures dropped below normal with local snow. The wind was weak to moderate. |
| April 12-13 2016 | Disturbances crossed the region on 9th, followed by a rain-unstable rise from 10th to 13th. |
| August 27 2016 | The weather was under some unstable intermissions, e.g., stormy on 27th and 28th, then a few showers remained on 29th. |
| October 25 2016 | With the gradual increase of pressure until 1035 hPa on 28th, low clouds and fogs were tenacious. |
| … | … |
| * Accessible at: https://donneespubliques.meteofrance.fr/?fond=produit&id_produit=129&id_rubrique=29 | |

[Figure]

Figure S3. (left) Time series of the observed and MYJ_BEP modeled hourly $CO_2$ concentration at SAC station from Jan 18th to 21st 2016. The grey shaded areas indicate the ranges of simulation results with five physical schemes used in this study (Table 1a in the manuscript). The yellow vertical lines indicate the large model-observation misfits (outliers) detected by the K-nearest neighbors (KNN) algorithm. (right) Distribution of the hourly $CO_2$ concentrations as a function of the wind speed for the year 2016.

[Figure]

Figure S4. (a) Hourly CO and $CO_2$ concentration measurements as a function of wind speed and direction at OVS station for the year 2016. (b) Image of the rooftop at OVS station with the CRDS CO & $CO_2$ sampling inlet in cyan and the building exhaust air system in red and green.

Page 7 line 2 – The authors say that individual measurement errors are negligible compared to model-data differences. However, model-data comparisons are made on hourly time scales and there will be (potentially large) variation within the hour. How is this sub-hour variation accounted for in model-data comparison?

In response to the reviewer's question about the sub-hour $CO_2$ variation, we also show the distribution of the standard deviation of all minute-scale $CO_2$ samples per hour at CDS station for the year 2016 (Figure R1). The median value of standard deviation is around 1.3 ppm and the upper quartile (75th percentile) is less than 2.5 ppm. The magnitude of this sub-hour variability is much smaller than the RMSE value (14.5 ppm, shown in Figure 4) of model-data comparison. We thus consider that the $CO_2$ measurements provided by the CRDS instrument have a high precision and the individual measurement errors are negligible compared to the model-data differences. Furthermore, the model was set to output concentration at hourly scale so that we do not have the sub-hourly simulated instant concentrations to make the high-frequency model-data comparison.

[Figure]

Figure R1. Distribution of the standard deviation of all minute-scale $CO_2$ samples per hour at CDS station for the year 2016. The midpoint, the box and the whiskers in the boxplot represent the 0.5 quantile, 0.25/0.75 quantiles, and 0.1/0.9 quantiles respectively. The median is shown as a horizontal line going through the box with the value in red.

Page 8 line 17 – The authors suggest upwind-downwind gradient can be used even in growing seasons as natural biogenic fluxes do not completely offset anthropogenic fluxes – but this still requires good knowledge of biogenic fluxes as they will make a major contribution to the observed mole fraction difference. The authors note several lines later that the biogenic fluxes show systematic errors, does this contradict the first statement?

The first statement intended to explain that biogenic fluxes in growing seasons do not offset the signal of anthropogenic emissions, but that uncertainty in biogenic fluxes makes it more difficult to quantify anthropogenic emissions in an inversion during the growing season. This is linked to the impact of biogenic fluxes on the knowledge of the anthropogenic contribution to the $CO_2$ gradients. We rewrote the two sentences in section 3.2.2 as follows:

The first statement as "During the afternoon, the $CO_2$ differences are mostly positive and result primarily from the larger contribution of the anthropogenic emissions at CDS, both during the growing and non-growing season. This result indicates that the magnitude of daytime net carbon uptake plants between the stations does not fully offset that of the anthropogenic emissions, and thus the $CO_2$ concentration gradients between the upwind and downwind stations that are used in previous inversion studies. Nevertheless, the biogenic contribution to the gradient is not negligible with a potential impact on the estimate of the anthropogenic emissions from the measured gradient."

The second statement as "The respiration emission at night seems to be underestimated by the VPRM model. If this nighttime respiration bias would be correlated with the daytime respiration bias (Reichstein et al., 2005), it would imply that modeled positive gradients of $CO_2$ between urban and rural stations could be overestimated during the growing season. We thus recommend for an inversion to control separately (with a priori) anthropogenic emissions and net ecosystem exchange, or even photosynthesis and respiration if additional data confirm a bias of respiration in VPRM."

Page 10 lines 1-21 (and Figure 11) – This section should be reworked for clarity. My understanding is that the authors have averaged the difference of tracer concentrations between runs across time (by month) and horizontal space (by land type) to calculate the individual values shows in figure 11 – but this should be made clearer in the writing. For many of the values, the standard deviation is large w.r.t. the mean difference. A different type of plot that shows the distribution, such as a violin plot, may be more appropriate.

Note: Figure 11 is now ranked as Figure 12 in the revised manuscript.

We have changed the legend of Figure 12, as well as the description at the beginning of the paragraph where Figure 12 is mentioned.

The modified legend is:

"Figure 12. Analysis of the $CO_2$ difference between the control run (BEP_MYJ) and each of the other four sensitivity runs over two one-month periods. The colored bars show the monthly mean difference whereas the black lines indicate +/- one standard deviation of the monthly values. The results are shown for two periods of the day (afternoon 11-16 UTC, nighttime 00-05 UTC) and for three land use types (urban, crop and the others)."

The modified text is:

"We also accessed the respective contributions of anthropogenic and biogenic fluxes to the simulated spread of $CO_2$ concentrations using different physics schemes. This allows an estimate of the impact of uncertainties in the atmospheric transport modeling along with that of the impact of the various flux contributions. Figure 12 shows the statistics of the differences in simulated anthropogenic and biogenic $CO_2$ at approximately 20 m AGL between the control run (BEP_MYJ) and each of the other four sensitivity runs."

Page 11 line 13 – A strong conclusion for the use of KNN outlier removal that is not justified, see above.

Please see the answer above.

Figure 12 – Why use a cumulative distribution and not a histogram – what are the authors trying to show with this choice?

Note: Figure 12 is now ranked as Figure 13 in the revised manuscript.

Figure 13b is changed to a basic histogram as suggested. The cumulative histogram makes it easy to quantify the fraction of data below or above a threshold, while the basic histogram is much easier for a visual representation of data distribution. Given that these two formats make no difference in representing the core findings of the study, we therefore changed it to a basic one to make the content more readable.

**Technical Comments**

Page 8 line 18 – suggest "(the SAC station had unfortunately measurement gaps)" -> "(unfortunately the SAC station had measurement gaps)" and dates of gap added.

Text is modified as suggested:

"(unfortunately the SAC station had measurement gaps from May 3rd to June 23rd and from July 7th to July 12th)"

General note on figures – Rainbow colour schemes should be replaced with perceptually uniform colour schemes, and red-green colour schemes should be avoided due to Colour vision deficiency (colour blindness).

We thank the reviewer for the suggestion. The rainbow colour scheme in Figure 8 has been replaced with a perceptually uniform colour scheme.

Figure 4 – Both the dark blue-filled contour and red contour are called the 'threshold' but the two are not in agreement (red contour seems to be the correct one).

Note: Figure 4 is now ranked as Figure 5 in the revised manuscript.

We thank the reviewer for having spotted the mistake. We have corrected the caption. The filled contour with the blue colormap is from the minimum anomaly score (dark blue) to a threshold value (light blue), so that it is the light blue and the red contour that indicates the threshold. The modified text is as follows:

"The shade of blue area indicates the anomaly score for each point, with the minimum in dark blue and the threshold value in light blue."

Figure 6 – A note to remind the reader that this figure is for January only would be helpful.

Note: Figure 6 is now ranked as Figure 7 in the revised manuscript.

We have added "in January" in the caption.

Figure 7 – This figure is dense, which hinders clarity. I suggest the wind direction and mf time series to be moved to new windows with a shared x axis. The boundary layer height should also have a larger contrast to make it more visible.

Note: Figure 7 is now ranked as Figure 8 in the revised manuscript.

Figure 8 is changed as suggested. The time series of the wind arrow and the model-data $CO_2$ concentration has been moved to a new panel (Figure 8b). Apart from changing the line color, we also added the PBL height measurements in Figure 8a.

Figure 10 – Showing the line of transect for the south-north slice on the lat-lon plots would make interpretation of the figure clearer.

Note: Figure 10 is now ranked as Figure 11 in the revised manuscript.

Figure 11 is changed as suggested. We have added a white dash line to show this south-north transect.

All references mentioned in this response are already included in the manuscript.

---

## Author Comment (AC2) · 29 Apr 2021

We would like to thank Referee #2 for the comments concerning our manuscript. Please find below a point-by-point response (in blue) to each of the comments raised by the reviewer (in black). All the figure numbers correspond to the revised manuscript.

**Anonymous Referee #2**

**General comments:**

The authors attempted to determine the uncertainty sources for the modeled $CO_2$ concentrations over Paris, France, using a set of WRF-Chem simulations varying with physics-based transport, fossil fuel emissions, and $CO_2$ boundary conditions, for 2016. They mainly focused on the impact of PBL schemes and with the combination of the urban canopy models, two fossil fuel emission inventories with /without hourly variability, and two global models as boundary conditions on the modeled $CO_2$ in comparison with the ground-based in-situ $CO_2$ measurements. Their results show that model-data mismatch maximizes in the nighttime so they recommended the readers to discard the model-data misfits and use afternoon measurements for inversion. This is not new, and I believe that is what we do in atmospheric inversion.

Regarding this specific item, the point here is the assessment of such a traditional practice in global to regional scale inversions to the specific, fast-growing and more recent field of inverse modeling i.e. that of urban $CO_2$ emissions based on ~1km resolution transport models, while the transport conditions and modeling skills over urban area can highly differ from larger scale transport conditions.

They also found the boundary condition could cause large differences at the synoptic scale and suggest exploring more about the influence of the boundary condition on the inversed results and suggest using additional observation to constrain boundary conditions. This is also not new. The authors are aware of these points because they cited those papers. So, I failed to locate the novelty of the work that brought into the community. The authors, in my opinion, have repeated some of the previous studies without extending the science further.

We thank the reviewer for giving us the opportunity to better explain the novelty of our results, and to better position our study with respect to similar one. In fact, few studies performed a detailed analysis of different sources of errors for modeling urban $CO_2$ similar to the one presented in our study (Table R1). Here we present original work for the city of Paris with a deeper analysis on the concept of assimilating cross-city gradients and evaluates the inversions strengths and weaknesses with the results from a full year worth of $CO_2$ measurements at 8 in-situ stations combined with the meteorological measurements, a sophisticated high-resolution atmospheric transport model coupled with the diagnostic biosphere VPRM model, and a series of sensitivity tests to the main components of the inverse modeling system.

The use of city downwind-upwind gradients for city-scale inversion has been tested for Paris and promoted by a series of few publications (Bréon et al., 2015, Staufer et al. 2016, Wu et al. 2016) cited in this paper. Although the obtained results demonstrate the effectiveness of the inversion system, there are also several aspects concerning its improvement. For instance, the study in Lac et al. (2013), among others (e.g., Kim et al., 2013), demonstrates the potential improvements in the meteorological and atmospheric $CO_2$ modeling over the Paris region. Therefore, it suggests investigating in more detail whether the urban effects on atmospheric transport modeling need to be accounted for in the inversion of $CO_2$ fluxes for Paris. Lian et al. (2018) and Lian et al. (2019) attempted at setting up a high-resolution atmospheric transport modeling framework that is more robust or at least more flexible in terms of parameterization than those used in the previous Paris studies to account for the impacts of the urban effects, the biogenic flux and the model

physics, which makes it promising to enlarge the set of data that can be assimilated for the inversions of the Paris $CO_2$ emissions, and in a more general way, to strengthen the inversions.

Moreover, since the publications by Bréon et al. (2015) and Staufer et al. (2016), the Paris $CO_2$ network has been expanded and relocated since the year 2014, with several new in-situ $CO_2$ stations combined with meteorological measurements. The present monitoring network, in particular the two newly built urban sites (JUS and CDS), is expected to provide new insights into the urban $CO_2$ characteristics and the high-resolution atmospheric $CO_2$ modeling. A full re-assessment of the modeling skill and of the main sources of misfits between the observations and the model was needed on these new bases. This study actually presents a much more extensive analysis of the source of errors in the simulation over a one-year long period, in particular of the meteo-transport modeling errors, and of the skill for simulating the data from the site in the core of the urban area, than in the previous publications. In addition, Paris is a megacity like Los Angeles but surrounded by much more active vegetation in the growing season. In this study, the biogenic $CO_2$ fluxes were calculated online in WRF-Chem by the diagnostic biosphere VPRM model at 1-km horizontal resolution. We have demonstrated the impact of the biogenic activity on night-time measurements, which was not done before.

In practice, even though the general conclusion converges towards those raised by the previous publications, the study conducted here provides some new error characterization and a range of new detailed insights on the signature of the different types of sources of errors at city scale during nighttime and daytime for the full year period. Our results not only reveal our greatest efforts and current ability to simulate the atmospheric $CO_2$ concentration in an urban environment, but also prepare a promising way for a better inversion of $CO_2$ emissions from Paris. Therefore, we believe that the experience gained on what can be done and not done over Paris, could provide useful insights to other cities.

We have modified the text in the introduction section accordingly:

"Since the year 2014, the Paris $CO_2$ monitoring network has been relocated and expanded with seven in-situ CO2 stations combined with meteorological measurements. The present network, in particular the two newly built urban sites, is expected to provide new insights into the urban $CO_2$ characteristics. Lian et al. (2018) and Lian et al. (2019) attempted at setting up a high-resolution atmospheric transport modeling framework that is more robust or at least more flexible in terms of parameterization than those used in the previous Paris studies to account for the impacts of the urban effects, the biogenic flux and the model physics, which makes it promising to enlarge the set of data that can be assimilated for the inversions of the Paris $CO_2$ emissions, and in a more general way, to strengthen the inversions. Therefore, a full re-assessment of the modeling skills and of the main sources of misfits between the observations and the model is needed on these new bases. More specifically, we analyze in detail the model-measurement mismatches so as to identify critical sources of errors that would compromise a high-resolution atmospheric inversion of urban $CO_2$ emissions in the Paris area. A set of forward simulations of atmospheric $CO_2$ concentration are performed at 1-km horizontal resolution using the WRF-Chem model (Grell et al., 2005) with different anthropogenic emission inventories, physical parameterizations and $CO_2$ boundary conditions over the Paris for the 1-year period spanning December 2015 to November 2016. The main objectives of this paper are to provide a rigorous and detailed error characterization of our atmospheric modeling system and to determine the data selection method (i.e. filtering of short-term model errors and local contamination) and $CO_2$ boundary condition specifications at city scale during both daytime and nighttime over the full year period. We also address the question to what extent these model-measurement mismatches might be reduced and how our proposed diagnostics could be used to provide additional constraints for the inversion of $CO_2$ emissions at the city scale."

Table R1. Few published studies with the objective to investigate the sources of error in atmospheric $CO_2$ modeling for city (comparison with this study).

| References | City | Objective | Study Period | Measurement | Note |
|---|---|---|---|---|---|
| Feng et al. 2016 | Los Angeles | Model-data comparison and network design evaluation | One month (mid-May to mid-June 2010) | $CO_2$ (2 in situ stations), PBL height, meteorological fields | Sensitivity test of physical scheme and spatial resolution |
| Martin et al. 2019 | Washington DC/Baltimore | Analysis of errors in transport and fossil fuel emission | One month (February 2016) | $CO_2$ (3 in situ stations + 1 rural station), PBL height, meteorological fields | Sensitivity test of four fossil fuel inventory |
| This study | Paris | Analysis of errors in fossil fuel emission, biogenic flux, atmospheric transport and $CO_2$ boundary condition | One year (December 2015 to November 2016) | $CO_2$ (6 in situ stations + 2 rural station), PBL height, meteorological fields | Sensitivity test of physical scheme, fossil fuel inventory and $CO_2$ boundary condition |

Besides, the authors cited a few pilot $CO_2$ urban studies, such as INFLUX and LA megacity. Both Lauvaux (2016) and Feng (2016) pointed out the significant improvement of using high resolution fossil fuel inventories in simulating $CO_2$ in the urban environment. Although the authors included two different fossil fuel inventories in the simulation with the variation of the temporal components, I have a hard time following the goal of the experimental design. I thought they would explore the sensitivity of modeled $CO_2$ to the temporal resolutions when I was reading the methods and section 3.2.1, but the related findings were not emphasized in the conclusion. Why?

This suggestion is well taken. We have added the following sentence in the method section to make the objective clearer:

"In order to investigate the impact of the spatio-temporal distribution (especially the prescribed diurnal profile) of emissions on the modeled $CO_2$ concentrations, we made a one-month simulation using these two anthropogenic inventories together with their respective temporal profiles (Table 1b). Within the same group of simulations, two more sensitivity tests of the diurnal profile were also carried out by using……"

We also add the related findings in the conclusion:

"Our results indicate that the temporal profile of the heating sector used by the AirParif inventory tends to bear a large uncertainty. It is one of the two major causes that led to the large model-data misfits during the nighttime. In the IdF region, $CO_2$ emissions from the heating sector are linked to the burning of gas and oil, and electricity consumption. We could expect that a more constant diurnal profile should probably be a better approximation to the truth than the current one. This hypothesis has been further justified by an independent analysis of daily gas use and hourly electric consumption data within the IdF region (unpublished analysis led by a co-author of this study, François Marie Bréon)."

One of the major concerns in the urban $CO_2$ studies falls in the impact of the biosphere around or within the cities. The results of this study also showed that the impact of the biogenic fluxes is significant and not negligible, meaning that the biosphere is another uncertainty source over Paris. The author can refer to Feng (2019a; 2019b) to construct a set of biospheric fluxes and investigate the uncertainty of the biospheric in

the modeled $CO_2$ as well. Additionally, the authors relied on the VPRM module in WRF-Chem to provide the biogenic fluxes. It's not clear to me that if VPRM has been tuned with flux towers or not. The VPRM parameters in WRF-Chem are fixed and needed to be tuned with the flux towers to have a relatively accurate biospheric flux estimation (Hilton et al., 2013: Hilton et al., 2014). If the authors used the default values for the VPRM parameters in this study, the authors will have to consider the errors caused by the biosphere during the interpretation of the results which is almost impossible to isolate from transport, emission, or boundary condition. However, because of the simplicity of VPRM, the authors can build a set of parameter-based perturbations of the biospheric fluxes via VPRM to address my first concern.

In this study, the VPRM parameters (α, β, λ, and PAR0) have been calibrated using the eddy covariance flux measurements for different vegetation types in Europe made during the Integrated Project CarboEurope-IP (http://www.carboeurope.org/).

More detailed descriptions in terms of the model setup used in this study, e.g. the domain setting, the nudging option and the VPRM model, can be found in Lian et al. (2019) paper. We have referred to this information at the beginning of section 2.1 in this manuscript:

"Details regarding the model setup and the reference data used in the simulations are outlined briefly below and described in Lian et al. (2019)."

For better clarity, we have also added the following statement in the revised manuscript:

"The values of the four parameters (α, β, λ, and PAR0) for each vegetation type used by VPRM have been optimized against eddy covariance flux measurements over Europe collected during the Integrated Project CarboEurope-IP (http://www.carboeurope.org/)."

Given the optimal VPRM parameters have already been implemented in the simulation, we thus feel that there is no need to make a set of user-defined parameter perturbations as suggested by the reviewer.

The authors chose five combinations of the PBL schemes and urban canopy models to study the impact of the transport and concluded that it's difficult to have "good" transport. First of all, what are the rationales the authors believe these two schemes are the key players of the $CO_2$ urban modeling? Díaz-Isaac (2018) using an ensemble approach pointed out PBL indeed is a major player but ranked No. 2. The most dominant parameterization is the land surface model used in the study. I am aware that the response of the modeled $CO_2$ to the model physics may vary when the location changes. Have the authors explored the impact of other model parameterizations on the simulated $CO_2$? This may be also why the model results have such a large bias in this study.

The experiment design and the selection of physical scheme in this study are mainly based on the results of our previous study (Lian et al., 2018). Lian et al. (2018) investigated for the city of Paris, whether the high-resolution WRF model with its various configurations, can provide a good representation of meteorological fields in support of tracer atmospheric transport modeling. A series of numerical experiments (32 simulations) were carried out with the goal of detecting the sensitivity of WRF to its various physics schemes (including 6 microphysics schemes, 3 radiation schemes, 5 cumulus convection schemes, 4 PBL & surface layer schemes, and urban canopy scheme) and nudging strategies (spectral nudging, grid nudging and the objective analysis program OBSGRID). The meteorology provided by WRF was evaluated against both ground-based and radiosonde vertical observations, with a focus on three atmospheric variables (air temperature, wind and PBL height) that are relevant to the $CO_2$ transport in an urban environment. Our sensitivity tests with different WRF physics schemes show that the wind speed and the PBL height are much more strongly influenced by PBL schemes with respect to other physics schemes. The WRF model

together with its urban canopy scheme makes it possible to represent the urban heat island effects. Results in Lian et al. (2018) show that the meteorological variables are generally well reproduced by our WRF setup with the objective analysis and multi-nudging options that have also been used in this study. Meanwhile, it also provides an objective method for us to select the appropriate model physical schemes. Thus in this study, we only carried out the sensitivity simulations with different PBL and urban canopy schemes as they are sufficient to address the paper main question regarding the ability of a configuration of the WRF-Chem model to simulate the atmospheric $CO_2$ transport over Paris, but also to provide an estimate of the atmospheric transport uncertainty. All the other physics options remained the same in the experiments.

In the manuscript, we already mentioned that:

"These options correspond to those selected by Lian et al. (2018) which showed good performances for simulating near-surface winds and temperatures over the Paris region."

For better clarity, we have also added the following sentence in the revised manuscript to account for the reviewer's comment:

"These two physics schemes were selected as they have a more significant impact on the simulated meteorological variables than the other schemes based on our previous sensitivity study (Lian et al., 2018, Lian et al., 2019), and thus the differences between simulations with these two physical options could provide an estimate of the atmospheric transport uncertainty over the Paris region."

Secondly, the model-data mismatches are extremely large and out of my expiation. For example, the whole year averaged diurnal mismatch can be as large as -10 to 5 ppm at the two urban sites in Figure 5. I found a similar figure in Figure 9 of Feng (2016), even though it's a month averaged value, in which the diurnal cycle from the high-resolution simulation looks almost identical to the observation. What causes the large bias in this work? I would check if any errors caused by other model physics.

Note: Figure 5 is now ranked as Figure 6 in the revised manuscript.

At first glance, the large model-data mismatches in Figure 6 shown in section "3.1 Overall model performance" might be a surprise since the model underestimates $CO_2$ with a bias ranging from 0 to 12 ppm across stations during the night until around 05 UTC. In fact, a fairly detailed explanation of the two causes of this model-data discrepancy was already provided and justified in section 3.2 of the manuscript. It is due to the prescribed nighttime heating emission profile used in the AirParif anthropogenic inventory (the second paragraph in section 3.2.1) and the nighttime model transport issue (the third paragraph in section 3.2.1).

Thirdly, the authors concluded that the transport issue is difficult to identify. I disagree. The model transport can be evaluated with meteorological observations. Apparently, there are meteorological observations at the monitoring sites. Additionally, there are quite a few WMO stations in the domain. Comparing with meteorological observations in the model domain will allow the authors to have a better sense of the model transport.

We certainly agree with the reviewer that the accuracy of the modeled $CO_2$ concentrations depends on the quality of the meteorological model. Therefore, as a first step, our previous Lian et al. (2018) paper particularly focuses on the evaluation of WRF in simulating the meteorological variables over the Paris region. The statistics for WRF results as compared to the observations show that our model setup for Paris with its multi-nudging options can well reproduce the near-surface air temperature and wind without obvious technical or configuration issues (Figure S2 in Lian et al. 2018). We thus feel that there is no need to fully resume such a detailed meteorological validation in this study. We acknowledge nevertheless that

the reader may want to see more so that we have provided the following assessment of the meteorological fields.

We have added a new sub-section (section 3.1.1 Meteorological fields) together with the two new figures (Figure 4 and Figure S2) in the revised manuscript.

"In this section, we start with an evaluation of the overall performance of the control run (BEP_MYJ) in simulating both meteorological fields and atmospheric $CO_2$ over the full-year period. Since the accuracy of the modeled $CO_2$ concentrations depends on the quality of the meteorological model, the simulated meteorology by WRF was first evaluated against observations at SAC100 and SIRTA stations with a focus on three variables (air temperature, wind and PBL height). Figure 4 shows the time series of the 1-year daily afternoon mean (11-16 UTC) observed and modeled temperature, wind speed and wind direction at SAC100 station, together with their statistics summarized in the scatter plots. The daily nighttime mean (21-05 UTC) data are shown in Figure S2. In general, both daytime and nighttime temperature are well reproduced by WRF with correlation coefficient, RMSE and MBE of 1.0, 0.44°C, 0.06°C and 0.99, 0.67°C, 0.23°C respectively. The analysis of the MBE shows that the wind speeds are slightly overestimated by WRF, with a bias of 0.96 m/s for afternoon and 0.68 m/s at night. As for the wind direction, the model-data misfits decrease with the increasing wind speed. Seasonal (and even some day-to-day) variations in the afternoon average PBL heights diagnosed from the model data are in general agreement with the observations at the suburban SIRTA site with a RMSE of 359 m and a positive bias of 82 m. Some disagreements between the model-data PBL height estimates can be expected given layer heights from aerosol-based methods (as here applied to the observations) tend to lag behind those determined from thermodynamic methods (applied to the model data) during the course of the day (Kotthaus et al., 2018). Relative agreement between PBL heights is reduced at night (Figure S2), as uncertainties are higher in both the observed layer heights (Section 2.2) and those diagnosed from the model data (Shin and Hong, 2011). In general, results in Figure 4 and Figure S2 show that the simulated meteorological fields agree reasonably well with observations both during day and night which indicates parameter settings suitable overall."

[Figure]

Figure 4. Time series of the daily afternoon mean (11-16 UTC) observed and BEP_MYJ modeled (a) temperature, (b) wind speed, (c) wind direction and (e) $CO_2$ concentration at SAC100 station. (d) Time series of the daily afternoon mean (11-16 UTC) observed and modeled PBL height at SIRTA station.

[Figure]

Figure S2. Time series of the daily nighttime mean (21-05 UTC) observed and BEP_MYJ modeled (a) temperature, (b) wind speed, (c) wind direction and (e) $CO_2$ concentration at SAC100 station. (d) Time series of the daily nighttime mean (21-05 UTC) observed and modeled PBL height at SIRTA station.

As the authors mentioned that boundary conditions can lead to large bias in the inversed results, the results showed that 5-20 ppm day-to-day difference between the two global models along the edges of the model domain. In the $CO_2$ regional (inverse) modeling, one of the major concerns about the boundary condition is the conservation of mass (Butler et al., 2020). How did the authors handle mass conservation when incorporating global modeled $CO_2$ into the regional model domain?

As described in the manuscript, both global datasets (CAMS and CarbonTracker) were interpolated onto the outermost domain of WRF-Chem (D01) (bilinearly in longitude, longitude and linearly in pressure) so as to provide the lateral boundary conditions for $CO_2$ simulations. We did not specifically address the pressure-weighted integrated columnar concentration of $CO_2$ as we only focus on the model-data comparison for the near surface in situ $CO_2$ measurements rather than the column-average dry-air mole

fractions of $CO_2$ (XCO2) measured by the satellite or the ground-based remote sensing system. In addition, it is worth pointing out that west winds (180-360° headings) are dominant in the Paris area. For most time of the year (~73%), the differences in simulated $CO_2$ concentrations over Paris are within the range of ±1 ppm since they are mainly affected by those differences between CAMS and CarbonTracker at the western boundary of D01. Even though the differences between CAMS and CarbonTracker are larger at the eastern boundary (-4.8 ± 7.4 ppm for 00 UTC and -1.7 ± 3.3 ppm for 12 UTC), the magnitude of uncertainties becomes much smaller after a long-distance transport of $CO_2$ (up to 5 ppm during several synoptic episodes). Under these circumstances, we also suggest the use of $CO_2$ gradients between upwind-downwind stations in the inversion for Paris, which will further decrease the impacts as compared to a mass-balance inversion method.

Another issue is that the number of boundary conditions used is too small to quantify the uncertainty. Strictly speaking, to be able to claim quantification of the uncertainty sources, a large number of the ensemble and a set of calibration procedures are required, such as rank histogram, reliability diagram, brier scores, etc. (Garaud and Mallet, 2011). Although it may be difficult to meet two criteria with the $CO_2$ modeling, the authors will at least need three of them to study the sensitivity.

We disagree on this point. By using the two state of the art global $CO_2$ atmospheric inversion products (CAMS and CarbonTracker), the results in this study do show a fairly detailed information of uncertainties linked with the boundary condition hypothesis. It also provides an insight for the use of $CO_2$ gradients between upwind-downwind stations in the inversion of $CO_2$ fluxes for Paris so as to remove the potential errors from the boundary conditions, in particular when winds blowing from the east during the period of inversion. It is worth noting that the sensitivity tests in this study do not intend to explore and cover the full uncertainty space. We are more interested in the order of magnitude with the current two realistic boundary conditions rather than the theoretical perturbations with three members (or even more). With this perspective, using these two most suitable products is well enough to achieve our objective.

In summary, this work claims that it has a quantitative evaluation of uncertainty sources in the $CO_2$ modeling, but the experimental design is far from achieving the goal. It eventually is merely a sensitive study of modeled $CO_2$ to the selected fossil fuel emissions, the combination of PBL and urban canopy models, and boundary conditions. The size of the ensemble they built does not allow them to do a solid quantification study. As I mentioned, this study appears repeating some of the previous studies without advancing the understanding the community already holds currently, neither in science nor in techniques.

It is well known that city-scale inversion bears a large number of challenges that we do not claim to solve at once. We believe that the present manuscript provides error characterization and a range of new detailed insights on the signature of the different types of sources of errors at city scale, that most likely will remain during the forthcoming years. Meanwhile, following the suggestion from Reviewer #3, we have modified the title to better reflect our intent.

There are no clear rationales why they made such selection as I pointed out with the transport "ensemble". The authors did not address the major issues in urban modeling, i.e., the impact of biosphere, and regional modeling, i.e., the conservation of mass when applying boundary conditions. They also failed to have a clear conclusion about the findings associated with fossil fuel emissions. In my opinion, this work is incomplete and must be extended to consider publication; these concerns I brought up can be addressed, which, however, will require a new design of the method. In addition to the specific comments I listed below, I would not recommend this MS to be a published in ACP.

Please see our answer above in terms of the VPRM biogenic flux, the conservation of mass, and the conclusion about the fossil fuel emissions.

**Specific comments:**

Section 2: There are important details missing in the description of the model setup.

1) Did the model use simulation cycles? If yes, how often is it? If yes, how was the $CO_2$ field addressed, initializing every time or being carried over simulation cycles?

Following the reviewer's suggestion, we have added the following sentence in section 2.1.3 to address this point:

"The simulation was restarted every 5 days with the $CO_2$ initial values from the previous run."

2) ERA-Interim and the outermost domain of WRF-Chem have quite different resolutions. What are the rationales that the authors used grid nudging over spectral nudging?

As mentioned above, the choice of the model configuration in this study corresponds to those selected by Lian et al. (2018) which showed good performances for a representation of meteorological fields in support of the atmospheric transport modeling. The impact of grid nudging, spectral nudging, and WRF OBSGRID program have been investigated in Lian et al. (2018). We used the combination of grid nudging, surface analysis nudging and observation nudging together with the objective analysis (the latter three are generated by OBSGRID) to maximize the benefit of assimilating surface and upper air meteorological observations. The model performance of this multi-nudging options (WRF_OA) was compared to the one with the spectral nudging (WRF_noOA). Results show that WRF_OA provides obvious improvements in modeled surface temperature and wind speed over WRF_noOA, and is therefore recommended to be used in the atmospheric transport modeling when an accurate description of reality is needed. More details regarding the nudging to different variables and their coefficients are all described in Lian et al. (2018).

We feel that there is no need to explain in detail why we used grid nudging instead of spectral nudging in this study for two reasons:

(1) Comparison of the performance of nudging techniques (e.g. grid nudging vs. spectral nudging) is not closely related to the objective of this study. It has already been investigated in our previous study and referred in this manuscript.
(2) The nudging strategy used in this study has already been described in section 2.1 as follows:
"The grid nudging option in WRF to relax the model to ERA-Interim on large scales was applied to temperature and wind fields at model levels above the planetary boundary layer (PBL) of the outer two domains. We also used the surface analysis nudging and observation nudging options to assimilate the National Centers for Environmental Prediction (NCEP) operational global upper-air (ds351.0) and surface (ds461.0) observation weather station data (https://rda.ucar.edu/datasets/ds351.0/; https://rda.ucar.edu/datasets/ds461.0/), which are described in more detail in Lian et al. (2018)."

3) As I mentioned in the general comments, have the VPRM parameters constrained with the flux tower measurements?

See answer above. In this study, the VPRM parameters have already been optimized with the eddy covariance flux measurements over Europe.

4) When the authors were incorporating $CO_2$ IC/BC to WRF-Chem, how did the author address the conservation of $CO_2$ mass?

See answer above.

5) When using global modeled $CO_2$ as IC, the discontinuity of the global and regional model dynamic can cause discrepancy of the $CO_2$ as well. How much the difference caused by the discontinuity would be?

We follow the traditional method used in the modeling community for the IC & BC interpolation and the WRF downscaling. The WRF outermost boundaries were set far away from the area of our interest, with three levels of nesting with horizontal grid spacing of 25, 5 and 1 km, covering Europe (D01), Northern France (D02) and the IdF region (D03) respectively. We thus believe that such a discontinuity of the global and regional model dynamic at the D01 boundaries is not a critical issue for the simulated $CO_2$ concentration over Paris, especially for the $CO_2$ gradients across the city. This paper did not aim at solving such specific secondary problems in depth, which is out of the scope of this study.

P 6, L 25-30: the author interoperated that the reason of the higher $CO_2$ concentrations in fall than in winter was due to the anticyclone keeping the high $CO_2$ in the domain for quite a while. I disagree. If it's due to meteorology, the impact on the fossil fuel $CO_2$ and biospheric $CO_2$ concentration should be the same. We should see lower $CO_2$ in the suburban sites, but we don't.

The figure below (Figure R2) and analysis of weather regimes show that the higher concentration in autumn 2016 are partly due to more anticyclonic events associated with $CO_2$ peaks from mid-October to mid-November 2016 compared to December 2015. The impact on the concentrations is not only a function of the atmospheric circulation, but also on the fluxes. Tracer simulations (Figure R2c and R2d) indicate that these peaks in autumn 2016 are mainly explained by a higher contribution of anthropogenic emissions compared to those of 2015, whereas the contribution of biogenic respiration is similar between autumn 2016 and winter 2015.

[Figure]

Figure R2. Time series of the BEP_MYJ simulated daily average (a) total, (b) background, (c) anthropogenic and (d) biogenic $CO_2$ concentration at CDS and SAC station

P7, L1-5: as I said earlier, the authors should be able to identify at least to some degree if the issues are in transport or boundary conditions by comparing with the meteo data.

See answer above.

P10, L10-15: what causes the different bias between the BEP and UCM schemes? I would like to see a deeper explanation of that instead of simply saying lower or higher.

Concisely, the two schemes differ in their representations of the near-surface mixing which leads to large differences in the modeled $CO_2$ concentrations. Figure S7a shows the annual average of the vertical distribution of $CO_2$ concentrations at JUS station for 24 hours a day for the two schemes BEP, UCM and their differences. It can be seen that both BEP and UCM schemes reproduce large vertical gradients in $CO_2$ concentrations in the low atmosphere levels, i.e. up to approximately 300 m AGL but mostly in the first 100 m. In general, the BEP scheme reproduces smaller $CO_2$ concentrations in the lower part of the atmosphere (<100m) than UCM does. After the sunrise at 5 UTC, the mixing by turbulent diffusion and afterwards by thermal plume dilute $CO_2$ to higher levels more quickly in BEP than UCM. The height of the boundary layer keeps increasing between sunrise and noon. During the afternoon, the boundary layer is well developed with enough mixing, resulting in $CO_2$ being transported to the upper layers through atmospheric convection. Figure S7b shows the vertical distributions of $CO_2$ concentrations during the afternoon (11-16 UTC) at JUS station for 12 calendar months. The UCM scheme reproduces a much larger vertical gradient in $CO_2$ concentrations close to the surface than the BEP scheme does, especially in the winter time. This is because of the high emissions (mainly household heating) and the more stable atmosphere in winter than in summer.

More precisely, the different depictions of the urban canopy parameters in the two modules (e.g. building heights, pervious area fractions, street canyons, heat capacity and thermal conductivity) and their impact on the energy budget and atmospheric transport are most likely to be the cause of the performance difference between BEP and UCM. The simulated near-surface $CO_2$ concentrations are highly sensitive to small differences in urban friction coefficient, wind velocity and vertical mixing associated with turbulence over the emission-rich areas (e.g. city center in winter). Even though we have modified the geometric and thermal parameters in the module over Paris based on the work of Kim et al. (2013), there are still many land surface characteristics that were not calibrated due to lack of detailed information of the urban form for the Paris city. A further improvement of the urban canopy schemes and a deeper analysis are out of the scope of this study.

Based on the discussion above, we have added the following sentence in the manuscript and Figure S7 in the supplement to account for the reviewer's comment:

"This is because the BEP scheme generates more mixing in the lowest atmosphere especially from 07 to 14 UTC in the day and in winter relative to summer, which reduces the vertical gradient and therefore the largest concentrations near the surface (Figure S7)."

[Figure]

Figure S7. (a) Annual average of the vertical distributions of $CO_2$ concentrations at JUS station for 24 hours of the day for BEP, UCM and their differences; (b) Vertical distributions of $CO_2$ concentrations during afternoon (11-16 UTC) at JUS station for 12 calendar months for BEP, UCM and their differences.

P11, L19-21: I agree that based on the current setup, there is little hope to improve the model performance. However, the authors can follow my suggestion listed in the general comments. For example, checking the land surface model used, comparing with meteo data, etc., to identify if the problem is caused by transport is the first step. Then the authors can look into the emission, boundary conditions, etc.

Please see our answers above in terms of the choice of physics schemes, the model-data validation for the meteorological variables. Our previous study and the subsequent analyses (section 3.1.1 together with Figure 4 and S2) in this study indicate that the model-data discrepancy is not merely due to an obvious error in the atmospheric transport modeling. We thus analyze in further detail these measurement-model discrepancies and attempt to identify cases when they appear to be mainly driven by uncertainties in the anthropogenic emissions, in the biogenic fluxes, in the physical parameterizations of the atmospheric transport model, or in the $CO_2$ boundary conditions, as presented in section 3.2.

Figure 5: please use local time in the x-axis instead of UTC. The much bigger issue is the large bias in the biases.

Note: Figure 5 is now ranked as Figure 6 in the revised manuscript.

The local time in Paris is one hour ahead of UTC (UTC+1) from November to March, and two hours ahead of UTC (UTC+2) from April to October. As the time zone difference is only one or two hours from UTC to the local time, it may not seriously affect our visual interpretation of the results. Moreover, given that the time scale for the other figures and text in this manuscript are all shown as UTC, we might prefer to use UTC in the x-axis of Figure 6. We have added the following text in the caption of Figure 2 to clarify this point:

"The local time in Paris is one hour ahead of UTC (UTC+1) from November to March, and two hours ahead of UTC (UTC+2) from April to October."

Please see our answer above in terms of the large bias.

The following references are used only in the responses to the reviewer's comments. All other references mentioned above are already included in the manuscript.

Kim, Y., Sartelet, K., Raut, J. C., and Chazette, P.: Evaluation of the Weather Research and Forecast/urban model over Greater Paris. Boundary-layer meteorology, 149(1): 105-132, 2013.

Lac, C., Donnelly, R. P., Masson, V., Pal, S., Riette, S., Donier, S., Queguiner, S., Tanguy, G., Ammoura, L., and Xueref-Remy, I.: $CO_2$ dispersion modelling over Paris region within the CO2-MEGAPARIS project, Atmospheric Chemistry and Physics, 13, 4941-4961, 2013.

Reference:

Butler, Martha P., Thomas Lauvaux, Sha Feng, Junjie Liu, Kevin W. Bowman, and Kenneth J. Davis. "Atmospheric Simulations of Total Column $CO_2$ Mole Fractions from Global to Mesoscale within the Carbon Monitoring System Flux Inversion Framework." Atmosphere 11, no. 8 (August 2020): 787. https://doi.org/10.3390/atmos11080787.

Díaz-Isaac, Liza I., Thomas Lauvaux, and Kenneth J. Davis. "Impact of Physical Parameterizations and Initial Conditions on Simulated Atmospheric Transport and $CO_2$ Mole Fractions in the US Midwest." Atmospheric Chemistry and Physics 18, no. 20 (October 16, 2018): 14813–35. https://doi.org/10.5194/acp-18-14813-2018.

Feng, S., Lauvaux, T., Newman, S., Rao, P., Ahmadov, R., Deng, A., et al. (2016). Los Angeles megacity: a high-resolution land–atmosphere modelling system for urban $CO_2$ emissions. Atmospheric Chemistry and Physics, 16(14), 9019–9045. https://doi.org/10.5194/acp-16-9019-2016

Feng, Sha, Thomas Lauvaux, Kenneth J. Davis, Klaus Keller, Yu Zhou, Christopher Williams, Andrew E. Schuh, Junjie Liu, and Ian Baker. "Seasonal Characteristics of Model Uncertainties From Biogenic Fluxes, Transport, and Large-Scale Boundary Inflow in Atmospheric $CO_2$ Simulations Over North America." Journal of Geophysical Research: Atmospheres 124, no. 24 (2019): 14325–46. https://doi.org/10.1029/2019JD031165.

Feng, Sha, Thomas Lauvaux, Klaus Keller, Kenneth J. Davis, Peter Rayner, Tomohiro Oda, and Kevin R. Gurney. "A Road Map for Improving the Treatment of Uncertainties in High Resolution Regional Carbon Flux Inverse Estimates." Geophysical Research Letters 46, no. 22 (2019): 13461–69. https://doi.org/10.1029/2019GL082987.

Garaud, D., and V. Mallet. "Automatic Calibration of an Ensemble for Uncertainty Estimation and Probabilistic Forecast: Application to Air Quality." Journal of Geophysical Research: Atmospheres 116, no. D19 (October 16, 2011). https://doi.org/10.1029/2011JD015780.

Hilton, T.W., K. J. Davis, and K. Keller. 2014. Evaluating terrestrial $CO_2$ flux diagnoses and uncertainties from a simple land surface model and its residuals, Biogeosciences, 11, 217-235, doi:10.5194/bg-11-217-2014.

Hilton, T.W., K. J. Davis, K. Keller, and N.M. Urban. 2013. Improving North American terrestrial $CO_2$ flux diagnosis using spatial structure in land surface model residuals, Biogeosciences, 10,4607–4625, doi:10.5194/bg-10-4607-2013.

---

## Author Comment (AC3) · 29 Apr 2021

We would like to thank the reviewer for the valuable comments and suggestions for improving our manuscript. Please find below a point-by-point response (in blue) to each of the comments raised by the reviewer (in black). All the figure numbers correspond to the revised manuscript.

**Anonymous Referee #3**

**General Comments:**

This study by Lian et al., 2020 attempts to identify and quantify significant sources of errors that can hinder the accurate estimation of urban-scale emissions. The objective of this study, as claimed by the authors, also includes demonstrating how these diagnostics can be used for inverse modelling studies. An ensemble of WRF-Chem simulations are performed, varying emission inventories (one month of simulations), PBL schemes and urban canopy schemes (one year of simulations), and boundary conditions (one year of simulations). The topic is fascinating and is essential to investigate the how sensitive is the emission estimate to the different components of the transport mechanisms (simulated by the model), flux variations, and assumptions/methods employed. This is a well-written manuscript with clearly described methods and results arranged in a logical order, which made the manuscript easy to follow. The conclusions drawn, based on their analyses, are reasonable and are applicable to those working on city-scale and mesoscale inversions of $CO_2$ using WRF-Chem.

We thank the reviewer for his work and this positive general assessment of the manuscript.

The study in this present form, however, fails to justify the title. Though the study considered an ensemble of simulations and subsequent analyses, it is still insufficient to make a quantitative estimation/evaluation of sources of errors in $CO_2$ simulations which is adequate to the broad spectrum of inverse modelling studies. I'd instead consider it as a study on diagnosing the effect of vertical mixing (PBL schemes), specific modelling criteria (urban schemes) as well as boundary conditions in city-scale modelling, in addition to assessing the sensitivity of simulations to the emission patterns. I believe that the title can be reworked accordingly to present the study appropriately. Additionally, some other sections require more clarification, modification, and further analysis, as mentioned below. Thus, I would recommend a major revision before considering for publication in ACP.

We greatly appreciate the reviewer's suggestion and fully agree that the title needs to be made more concise and to have a clearer description of the objective so that the reader can more easily understand the analysis undertaken. We thus have altered the title to better reflect our intent:

"Sensitivity to the sources of uncertainties in the modeling of atmospheric $CO_2$ concentration within and in the vicinity of Paris city"

Though it is mentioned as one of the objectives, I don't see that the study has addressed the question to what extent or how the model-measurement error can be reduced. This is a major concern of mine. I'd consider that the authors could devise efficient analysis strategies to address this, given the availability of measurements from 6 +2 sites and ensemble of simulations. An adequate diagnosis of model-measurement mismatches is missing, which is a weakness of this manuscript. A discussion on how the diagnostic results can be used for the betterment of inversion studies is vaguely articulated in the manuscript. All the guidelines for the data selection put forward by the study (such as discard data with high model-data misfits, use only afternoon values; test the influence of boundary conditions) is already known to the community and currently practised in inverse model calculations. I would suggest authors avoid the above statement of

objective or revise thoroughly while incorporating additional analysis to explain the novelty of their findings.

We have expanded the discussion section and attempted to use it to consolidate the conclusions as far as possible. The following two paragraphs have been added to provide the perspective on reducing the model-measurement error that would be beneficial to the future atmospheric inversion studies.

"Furthermore, it remains difficult to interpret and use quantitatively in situ measurements within the city as long as there is no proper information about the turbulent airflow within and above the urban canopy. The near-surface mixing is not only controlled by the atmospheric stability conditions but also affected by the urban roughness and anthropogenic heat production. If the complex vertical mixing processes cannot be properly constrained in the transport model, it will be difficult to use the measurements acquired close to the sources in the atmospheric inversion system. Therefore, regular measurements of vertical $CO_2$ profiles, combined with relevant upper-air meteorological data (e.g., potential temperature and wind) and the mixing layer heights in the lower troposphere are expected to be included in the future Parisian $CO_2$ monitoring network. Such complementary measurements will be of great help to understand the characteristics of $CO_2$ vertical distribution under both stable and convective boundary-layer conditions. It can also be used to verify and validate the atmospheric transport model, and to reduce transport errors based on the data assimilation of more meteorological observations, leading to much higher accuracy in the atmospheric inversion system that aims at retrieving urban $CO_2$ fluxes."

"Focusing on the Paris region, two limitations of this study should be acknowledged and worth further investigating based on the high-resolution urban ecosystem modeling and monitoring so as to better quantify the impact of urban biogenic fluxes: (i) due to the coarse-resolution (1 km) SYNMAP land use data used for the VPRM model, the simulated biogenic fluxes in center Paris in this study are almost zero except for a few grid cells containing two big parks that are located in the eastern and western outskirts of the Paris city. While in reality, there are still a number of green space and pervious landscaped areas unevenly distributed in the city of Paris that need to be considered with a fine-scale (sub-kilometer) model; (ii) there is a lack of validation of the Paris-VPRM model in this study since no eddy covariance measurement is available within the Paris urban area and its surroundings. This limitation could be overcome by an expansion of the observation network with the neighborhood-scale urban eddy covariance flux measurements included."

**Specific Comments:**

Fig. 2. For Line style descriptions, please use another colour (e.g. black) than those used as line colours. Blue is already used for "Total". I had a hard time understanding. Also, I'd suggest you remove (c) and (d) and include AirParif (daily) and Constant in (a) and (b).

Figure 2 is changed as suggested.

Differences between CAMS and CarbonTracker at the four lateral boundaries: Why are there substantial differences between daytime and nighttime in East boundaries, sometimes even in opposite phases (Fig. 3b)? Please explain.

The CAMS and CarbonTracker could be considered as two independent global $CO_2$ analyses and atmospheric inversion systems. They have used a variety of different inputs (e.g. different prior estimates of $CO_2$ fluxes, observation datasets, etc.) and approaches or tools (e.g. different global atmospheric transport models with different meteorological forcing, different data assimilation methods with from the variational method applied to long temporal window and an ensemble Kalman filter-based method applied

to much shorter analysis windows) to estimate the $CO_2$ surface fluxes and transport them to simulate the three-dimensional atmospheric $CO_2$ mole fraction. Therefore, it is hard to fully rule out the detailed causes of the substantial difference at the eastern boundary. A possible explanation is the larger differences between both the fossil fuel and biogenic $CO_2$ fluxes over the European continent as a result of their respective analysis (http://www.globalcarbonatlas.org/en/CO2-emissions). It may also because of the sensitivity of the modeled $CO_2$ concentrations to the transport fields over the mountain region in the eastern part of the outer domain.

We have added the following sentence in the manuscript to account for the reviewer's comment:

"A possible explanation could be that both fossil fuel and biogenic $CO_2$ fluxes and associated uncertainties are larger over the European continent than over the oceans. It may also be caused by the sensitivity of the modeled $CO_2$ concentrations to the transport fields over the Alps mountain region at the eastern boundary."

Evaluation with in situ observations: I am not very convinced with the usage of KNN method? How is the outlier fraction calculated? How sensitive is the filter size in another outlier fraction? What are the criteria for the choice of 0.1 in this case?

Regarding the evaluation in section 3.1, it is worth noting that we also look at statistics without removing outliers, and the text is based on the analyses both with and without the KNN method. Moreover, please see our answer to Reviewer #1 on this topic where we show indications that this method successfully removes outliers that are due to the measurement contaminations from local unresolved sources of emissions and/or the model's inability under complex meteorological conditions. Regarding the conclusion/discussion section, we recognize that it would be more appropriate to encourage the use of this KNN algorithm based on a deeper analysis of the detected outliers instead of using it as a crude data filtering. We thus have rephrased the text in the revised manuscript (also see answer to Reviewer #1 for details).

The basic idea is that errors from emissions or from patterns that can be modeled at 1 km resolution should have a frequency and timescale longer than that of misfits removed by the algorithm. The algorithm is expected to remove isolated outliers that are not representative of normal conditions. It remains an arbitrary issue in terms of the definition of an outlier and the decision of whether to remove or keep it.

In response to the reviewer's concerns, we made a test of sensitivity to the outlier fraction, with values for this fraction ranging from 0 to 0.3. Figure R3 shows the change of the RMSE and MBE as a function of the outlier fraction values at one urban station (CDS) and one suburb station (SAC). In this study, the choice of 0.1 was an arbitrary selection based on the visual judgment of Figure R3 and Figure 5 (scatter plot of the model-data comparison).

[Figure]

Figure R3. Change of the RMSE and MBE (statistics for all hourly model-data $CO_2$ comparison from December 2015 to November 2016) as a function of the outlier fraction values used in the KNN method at one urban station (CDS) and one suburb station (SAC).

Given that outliers are removed, why the model-observation mismatch is this high (Fig. 4)? How can these mismatches be reduced? How about background sites in terms of evaluation? In addition to reporting the mismatches, I'd suggest the authors explain the reasons for these large deviations from observations. This is critical as I also see unexpectedly significant model-measurement differences in diurnal averages. Have authors checked different choices of physics/dynamics schemes or other parameters available in WRF-Chem to reduce this mismatch?

Note: Figure 4 is now ranked as Figure 5 in the revised manuscript.

This large all hourly model-data mismatch is mainly because the model underestimates $CO_2$ with a bias ranging from 0 to 12 ppm across stations during the night until around 05 UTC (also see Figure 6). In fact, a fairly detailed explanation of the two main causes of this model-data discrepancy was already provided and justified in section 3.2 of the manuscript. It is probably due to the prescribed nighttime heating emission profile used in the AirParif anthropogenic inventory (the second paragraph in section 3.2.1) and the nighttime model transport issue (the third paragraph in section 3.2.1). More precisely, our results first indicate that the temporal profile of the heating sector used by the AirParif inventory (with a significant decrease along the night) tends to bear a large uncertainty. In the IdF region, $CO_2$ emissions from the heating sector are linked to the burning of gas and oil, and electricity consumption. We could expect that a more constant diurnal profile should probably be a better approximation to the truth than the current one. This uncertainty could be justified and reduced by a further analysis of these related source data. Moreover, another two paragraphs in terms of the atmospheric transport and the modeled biogenic flux have also been added in the discussion section to provide the perspective on reducing the model-measurement error (please see our answer above).

We have also checked the background site at TRN station, located 101 km away from the center of Paris (see Figure 1) with air inlets placed at 5m, 50m, 100m and 180m AGL respectively. Figure R4a shows the scatter plot of the observed and BEP_MYJ (control run) simulated all hourly $CO_2$ concentrations at 50m and 100m sampling heights from December 2015 to November 2016. The comparison of the average diurnal variations between observation and model is shown in Figure R4b. Results show that the model also underestimates the $CO_2$ concentrations at this background site during the night, which is consistent with the

conclusions obtained from the stations within the IdF region. It is worth noting that both the observation and the model show an increase of $CO_2$ during the night at TRN, whereas the opposite was found to be true for the modeled value within IdF. This is because the fossil fuel $CO_2$ emissions outside the IdF region were taken from the IER inventory that has a more constant nighttime diurnal profile than the AirParif does.

We also made the model-data $CO_2$ comparisons of the other four sensitivity tests of physics schemes in this study (Table 1a). The results are shown to be similar to the control run (BEP_MYJ), except that the model runs with the UCM scheme tend to have larger misfits at the two urban stations (JUS and CDS). The two schemes differ in their representations of the near-surface mixing which leads to large differences in the modeled $CO_2$ concentrations (please see our answer to Reviewer #2 for details). On the other hand, we also evaluated WRF simulated meteorological fields against the observations. Results indicate that the model does not suffer from obviously inappropriate/wrong parameter settings that could lead to a significant transport error (also see our answer to Reviewer #2 for details).

[Figure]

Figure R4. (a) Observed and BEP_MYJ (control run) simulated all hourly $CO_2$ concentrations at TRN site with sampling heights at 50m and 100m above ground level respectively from December 2015 to November 2016. (b) Comparison of the average diurnal variations between the observed (dash line) and modeled (solid line) $CO_2$ concentrations, and the model-data $CO_2$ misfits at TRN site.

Sect. 3.2.1 "modeled value (green line in Figure 6) gets somewhat closer to the observation" In Fig. 6, the blue curve represents constant emissions. Please check. The simulations (BEP_MYJ_CON) reproduce the observed patterns better than other simulations; however, not in magnitude. So please rephrase the sentence accordingly: "modeled value (green line in Figure 6) gets somewhat closer to the observation".

Note: Figure 6 is now ranked as Figure 7 in the revised manuscript.

As described in section 2.1.2, the green line (BEP_MYJ_AIP) is the AirParif inventory with a constant temporal profile (each pixel has a different emission, but constant in time based on the temporal average of the AirParif inventory). The blue line (BEP_MYJ_CON) is a constant and spatially homogeneous emission where the emissions are distributed uniformly over the IdF whole territory.

The original sentence in section 3.2.1 is "Indeed, when assuming an emission constant in time, the decreasing trend is reduced and the modeled value (green line in Figure 7) gets somewhat closer to the observation."

For better clarity, we have changed the "an emission constant in time" to "the AirParif inventory with a constant temporal profile". The modified text is as follows:

"Instead, the decrease of anthropogenic emissions during the night (Figure 2) explains part of the decrease in modelled concentrations. Assuming the AirParif inventory with a constant temporal profile, the decreasing trend at night is reduced and the modeled value (green line in Figure 7) is closer to the observation than the control run (BEP_MYJ)."

Also please indicate the season (or January) in the Figure 6 caption.

Note: Figure 7 is now ranked as Figure 8 in the revised manuscript.

Done

Also, I am happy to note that the authors demonstrate the effect of emission trend and atmospheric vertical mixing here. Please comment on the effect of boundary conditions (though I expect it to be minimal by looking at the patterns in BEP_MYJ_CON).

We have updated Figure 7 to include the average diurnal cycle for the biogenic and background $CO_2$ along with the one of the total and anthropogenic $CO_2$ concentrations. It is worth noting that only the control run (BEP_MYJ) was plotted. This is because there were 6 distinct $CO_2$ tracers within this one-month simulation. Four of them corresponded to each of the four sensitivity tests of anthropogenic emissions to record their distinct atmospheric $CO_2$ signatures. The other two tracers were for the biogenic and background $CO_2$ respectively. Therefore, there will be no difference in the simulated biogenic and background $CO_2$ values among the four sensitivity tests. Results in Figure 7c and 7d show that the impacts of biogenic flux and background condition on this simulated decrease are relatively small, which is on the order of a fraction of a ppm.

The revised manuscript has included the following sentence as suggested by the reviewer:

"Results in Figure 7c and 7d show that the impacts of biogenic flux and background condition on this simulated decrease are relatively small as they are on the order of a fraction of a ppm."

PBLH and vertical distribution of the modelled $CO_2$ (BEP_MYJ): It is not clear to me why authors have mentioned Nielsen-Gammon et al., 2008 for PBLH estimation. What extent the MYJ scheme and Nielsen-Gammon et al., 2008 differ in deriving the PBLH? If different, a comparison plot will be helpful here.

Numerous thermodynamic parameters, including temperature, humidity and their derivatives (e.g., potential/virtual potential temperature) have been widely used to define the PBL height. The 1.5-theta-increase method defines boundary layer top based on the potential temperature and it has widely proven to be fairly representative and practical for PBL height detection. The diagnosed PBL heights obtained directly from different PBL schemes in WRF are calculated based on their own specific formulations. The MYJ scheme determines the PBL height using the turbulent kinetic energy (TKE) profile. The top of the PBL is defined to the height where the TKE profile decreases to a prescribed low threshold value (Janjic 2001). For the purpose of a fair comparison among the 3 PBL schemes (YSU, MYJ, BouLac) tested in this study, the 1.5-theta-increase method was used as a criterion to diagnose PBL heights. However, these comparison results are not shown in Figure 8 as they might be fruitless for the objective here.

In response to the reviewer's concerns, we performed a comparison of PBL heights estimated by the model against measurements at SIRTA station located about 20 km southwest of Paris center during January 2016 (Figure R5). Generally, the temporal variation of the diagnosed PBL heights is shown to be similar for the MYJ method and the 1.5-theta-increase (NG) method. The correlations with hourly observations are similar for MYJ (0.57) and NG (0.56), but the associated RMSE and MBE are better for NG. Both methods indicate that the model control run (BEP-MYJ) overestimates the PBL heights with a MBE of 136 m for MYJ and 54 m for NG.

[Figure]

Figure R5. Time series of the BEP-MYJ (control run) simulated PBL heights diagnosed by the MYJ scheme (in blue) and the 1.5-theta-increase method (NG, in green) with respect to the observations at SIRTA station (in red) for January 2016.

I am a bit surprised with the high PBL values in winter (initial half-month) over Paris. Do authors look at the PBL measurements (e.g. lidar measurements)? Fig. 7 is confusing as the 34-m AGL curves have nothing to do with the left Y-axis values. I would suggest authors separate these two curves (magenta and pink) from this and make an independent plot along with PBLH.

Note: Figure 7 is now ranked as Figure 8 in the revised manuscript.

Figure 8 has been modified as suggested. The time series of the wind arrow and the model-data $CO_2$ concentration has been moved to a new panel (Figure 8b). Apart from changing the line color, we also added the PBL height measurements in Figure 8a. The PBL heights data were obtained at the SIRTA station located about 20 km southwest of Paris center (Kotthaus et al., 2020. https://sirta.ipsl.fr/) as shown in Figure 1 in the revised manuscript. The comparison of all hourly measured and modeled PBL heights show that the model could well reproduce the temporal variation of the PBL heights in January and there is not an obvious long-term bias.

In addition, we also carried out a one-year validation of the PBL heights. Results in Figure 4d and Figure S2d further confirm that the model is capable of reproducing the PBL heights for other months in the year.

Sect. 4: Please see my comments above (w.r.t title) and refine this section thoroughly.

Please see the answer above.

**Minor comments:**

Page 7, "from 18 pm to 22 pm)" Please change to 18:00 UTC to 22:00 UTC.

Text is changed as suggested.

All references mentioned in this response are already included in the manuscript.

---

## Author Response (AR2)

**Comments to the Author:**

The authors have made a set of substantial additional analyses to improve the article, which are well represented within the scope of the manuscript. They addressed most of the reviewer comments adequately and proposed changes accordingly in their revised article. Also included explicit statements to support the significance of the study, which were missing in the earlier version. The title of the manuscript is changed to best match the content of the manuscript.

**Requested revision: clarify the following issue.**

Fig.4 in acp-2020-540-AC2-supplement.pdf:

Panels (a and b): The model's performance (temperature and wind speed) is too GOOD here (even R becomes 1 in (a))!

Is it because of data assimilation/nudging as explained in Sect. 2.1 (preprint version)?

["We also used the surface analysis nudging and observation nudging options to assimilate the National Centers for Environmental Prediction (NCEP) operational global upper-air (ds351.0) and surface (ds461.0) observation weather station data ...."]

If authors use the data from these sites for data assimilation, it is not proper to present Fig. 4a-c(+Fig.S2a-c) as part of the model evaluation.

Figure R1 shows the location of the World Meteorological Organization (WMO) surface and upper-air meteorological stations over Paris contained in the data assimilation process of the NCEP operational global weather observation subsets used in the WRF-OBSGRID program (https://rda.ucar.edu/datasets/ds461.0/#metadata/detailed.html?_do=y). These stations are operated by Météo-France, the official meteorology and climatology service in France. The assimilated meteorological observations include: i) the measurements of 2 m temperature, 10 m wind and moisture fields from seven surface stations, ii) the radiosondes at 12-houly intervals at the Trappes station.

**The meteorological data at the SAC station** used in the model-observation comparison (Fig.4 and Fig.S2) **are not included in the data assimilation/nudging process.** The SAC tall tower is operated by Laboratoire des Sciences du Climat et de l'Environnement (LSCE), CEA/CNRS/OVSQ for the purpose of conducting scientific research. This station is equipped with high-precision $CO_2/CO/CH_4$ analyzers with two air inlets placed at 15 and 100 m above ground level respectively. It is part of the ICOS atmosphere network measuring the atmospheric greenhouse gas concentrations in Europe. The SAC station is also equipped with meteorological instruments at 10, 40, 60, 80 and 100 m above ground level to measure various meteorological variables, e.g. air temperature, wind speed and wind direction, humidity. These data sets are available upon request, but are not included in the WMO global network. Similar as the SAC station, the CDS $CO_2$ monitoring station is also complemented by continuous meteorological measurements at 34 m above ground level.

The statistics for model-observation comparison of temperature and wind at SAC station are good. This may be due to the fact that: i) SAC is a suburb station and little influenced by the urban effects, ii) the measurements are taken at 100 m above ground level, iii) The correlation coefficient for temperature is also driven by the annual and seasonal variations that are accurately constrained in the model. In order to address the reviewer concern, we also made a model-observation comparison

at the CDS station which is located at the center of Paris (Figure R2). The meteorological instruments for wind and temperature at CDS were installed in January and March 2016 respectively, so that no observations are available at the beginning of the year. As expected, the performance for the modeled meteorological fields in urban areas is not as good as that in suburbs. The analysis of the MBE shows that the wind speeds are overestimated by WRF, with a bias of 1.49 m/s for afternoon and 0.97 m/s at night at CDS. The wind speed bias at SAC is 0.96 m/s for afternoon and 0.68 m/s at night. The correlation coefficient for temperature at CDS is also at a good score (0.99), but the values of RMSE (1.42°C) and MBE (-1.15°C) are inferior to those at SAC (0.44 °C and 0.06 °C) during the daytime.

In general, the simulated daytime and nighttime meteorological fields agree reasonably well with observations both for the urban and for the suburb areas. Given that the model-data validation at CDS brings a similar conclusion on the good model performance as the diagnostics made at SAC, we feel that there is no need to add Figure R2 in the revised manuscript.

We have added the following text in section 2.2 in the revised manuscript:

"Let us remind that the meteorological data at the SAC station are not included in the data assimilation process of the NCEP operational global weather observation subsets used in the WRF nudging program (section 2.1). They could therefore be considered as independent observations for the evaluation of the model performance in simulating the meteorological fields."

[Figure]

Figure R1. Distribution of the CO2 in situ and meteorological stations. The grey areas are the urban and build-up areas drawn from the CORINE land cover dataset at its native resolution of 250 m. The red dots are the WMO surface and upper-air meteorological stations contained in the data assimilation process of the NCEP operational global weather observation subsets used in the WRF-OBSGRID program. The blue triangles are the CO2 in-situ stations. The SAC station and CDS station are also equipped with meteorological instruments to measure the air temperature, wind speed and wind direction.

[Figure]

Figure R2. Time series of the daily (a) afternoon mean (11-16 UTC) and (b) nighttime mean (21-05 UTC) observed and BEP_MYJ modeled temperature, wind speed and wind direction at CDS station.

---

## Author Response (AR3)

Comments to the Author:

You mention the high correlation of the SAC data with that of the nearby stations which are included in the data assimilation. I think you need to be careful saying they are 'independent' observations, as they are not independent from a strict statistical perspective.

You could try this from line 23 on page 6:

"In addition to the $CO_2$ measurements, the hourly air temperature, wind speed and wind direction are measured at a height of 100 meters above ground level at the SAC station. The meteorological data from SAC are not included in the data assimilation process of the NCEP operational global weather observation subsets used in the WRF nudging program (section 2.1). We therefore use them in the evaluation of the model performance in simulating the meteorological fields."

Text changed as suggested. Thank you.